# SAGMAN: Stability Analysis of Graph Neural Networks on the Manifolds

## Abstract

Modern graph neural networks (GNNs) can be sensitive to changes in the input graph structure and node features, potentially resulting in unpredictable behavior and degraded performance. In this work, we introduce a spectral framework known as SAGMAN for examining the stability of GNNs. This framework assesses the distance distortions that arise from the nonlinear mappings of GNNs between the input and output manifolds: when two nearby nodes on the input manifold are mapped (through a GNN model) to two distant ones on the output manifold, it implies a large distance distortion and thus a poor GNN stability. We propose a distance-preserving graph dimension reduction (GDR) approach that utilizes spectral graph embedding and probabilistic graphical models (PGMs) to create low-dimensional input/output graph-based manifolds for meaningful stability analysis. Our empirical evaluations show that SAGMAN effectively assesses the stability of each node when subjected to various edge or feature perturbations, offering a scalable approach for evaluating the stability of GNNs, extending to applications within recommendation systems. Furthermore, we illustrate its utility in downstream tasks, notably in enhancing GNN stability and facilitating adversarial targeted attacks.

## 1 Introduction

The advent of Graph Neural Networks (GNNs) has sparked a significant shift in machine learning (ML), particularly in the realm of graph-structured data (Keisler, 2022; Hu et al., 2020; Kipf & Welling, 2016; Veličković et al., 2017; Zhou et al., 2020). By seamlessly integrating graph structure and node features, GNNs yield low-dimensional embedding vectors that maximally preserve the graph structural information (Grover & Leskovec, 2016). Such networks have been successfully deployed in a broad spectrum of real-world applications, including but not limited to recommendation systems (Fan et al., 2019), traffic flow prediction (Yu et al., 2017), chip placement (Mirhoseini et al., 2021), and social network analysis (Ying et al., 2018). However, the enduring challenge in the deployment of GNNs pertains to their stability, especially when subjected to perturbations in the graph structure (Sun et al., 2020; Jin et al., 2020; Xu et al., 2019). Recent studies suggest that even minor alterations to the graph structure (encompassing the addition, removal, or rearrangement of edges) can have a pronounced impact on the performance of GNNs (Zügner et al., 2018; Xu et al., 2019). This phenomenon is particularly prominent in tasks such as node classification (Yao et al., 2019; Veličković et al., 2017; Bojchevski & Günnemann, 2019). The concept of stability here transcends mere resistance to adversarial attacks, encompassing the network's ability to maintain consistent performance despite inevitable variations in the input data (graph structure and node features).

In the literature, while there are studies primarily focused on developing more stable GNN architectures (Wu et al., 2023; Zhao et al., 2024; Song et al., 2022; Gravina et al., 2022), a few attempts to analyze GNN stability comprehensively. Specifically, (Keriven et al., 2020) first studied the stability of graph convolutional networks (GCN) on random graphs under small deformation. Later, (Gama et al., 2020) and (Kenlay et al., 2021) explored the robustness of various graph filters, which are then used to measure the stabilities of the corresponding (spectral-based) GNNs. However, these prior methods are limited to either synthetic graphs or specific GNN models. Recent survey papers, such as (Dai et al., 2024), highlight the critical role of stability analysis in trustworthy GNNs.

In this work, we present SAGMAN, a novel framework devised to quantify the stability of GNNs through individual nodes. This is accomplished by assessing the resistance-distance distortions incurred by the nonlinear map (GNN model) between low-dimensional input and output graph-based manifolds. It is crucial to note that this study aims to offer significant insights for understanding and enhancing the stability of GNNs. While most robustness techniques focus on enhancing overall model robustness through architectural modifications or training strategies, they often lack fine-grained, node-level stability assessments. In contrast, SAGMAN introduces a spectral framework that quantifies stability at the individual node level, enabling targeted interventions such as precise attacks or tailored stability enhancements. The key technical contributions of this work are outlined below:

• This study introduces a spectral framework (SAGMAN) for measuring the stability of GNN models at the node level. This is achieved by effectively assessing the distance distortions caused by the maps between the input and output smooth manifolds.

• To construct the input smooth manifold for stability analysis in SAGMAN, we introduce a nonlinear graph dimension reduction (GDR) framework to transform the original input graph (with node features) into a low-dimensional graph-based manifold that can well preserve the original graph's spectral properties, such as effective resistance distances between nodes.

• SAGMAN has been empirically evaluated and shown to be effective in assessing the stability of individual nodes across various GNN models in realistic graph datasets. Moreover, SAGMAN allows for more powerful adversarial targeting attacks and greatly improving the stability (robustness) of the GNNs.

• SAGMAN has a near-linear time complexity and its data-centric nature allows it to operate across various GNN variants, independent of label information, network architecture, and learned parameters, demonstrating its wide applicability.

## 2 BACKGROUND

### 2.1 STABILITY ANALYSIS OF ML MODELS ON THE MANIFOLDS

The stability of a machine learning (ML) model refers to its ability to produce consistent outputs despite small variations or noise in the input data (Szegedy et al., 2013). To assess this stability, we utilize the *Distance Mapping Distortion* (DMD) metric (Cheng et al., 2021). For two input data samples $p$ and $q$, the DMD metric $\delta^M(p, q)$ is defined as the ratio of their distance on the output manifold to the one on the input manifold:

$$\delta^M(p, q) \overset{\text{def}}{=} \frac{d_Y(p, q)}{d_X(p, q)}. \tag{1}$$

By evaluating $\delta^M(p, q)$ for each pair of data samples, we can assess the stability of the ML model. Specifically, if two nearby data samples on the input manifold are mapped to distant points on the output manifold, this indicates a large $\delta^M(p, q)$ or equivalently a large local Lipschitz constant, and thus poor stability of the model near these samples; On the other hand, a small $\delta^M(p, q)$ implies that the model is stable in that region.

### 2.2 PREVIOUS INVESTIGATIONS ON THE STABILITY OF GNNS

The stability of a GNN refers to its output stability in the presence of edge/node perturbations (Sun et al., 2020). This includes maintaining the fidelity of predictions and outcomes when subjected to changes such as edge alterations or feature attacks. A desired GNN model is expected to exhibit good stability, wherein every predicted output or the graph embeddings do not change drastically in response to the aforementioned minor perturbations (Jin et al., 2020; Zhu et al., 2019). Several recent studies have underlined the importance of analyzing the stability of GNNs. For instance, Sharma et al. (2023) examined the task and model-agnostic vulnerabilities of GNNs, demonstrating that these networks remain susceptible to adversarial perturbations regardless of the specific downstream tasks. Furthermore, Huang et al. (2023) investigated robust graph representation learning via predictive coding, proposing a method that enhances GNN stability by reconstructing input data to mitigate the effects of adversarial attacks.

Moreover, while recent studies have investigated the stability issues of Graph Neural Networks (GNNs) on synthetic graphs or specific models (Keriven et al., 2020; Kenlay et al., 2021), they have not provided a unified framework for evaluating GNN stability. The latest state-of-the-art method (Li et al., 2024) develops such a unified framework but does not consider feature perturbations. This omission leaves a significant gap in understanding how GNNs respond to changes in node features, which is crucial for applications where feature data may be noisy or subject to perturbations.

## 3 THE SAGMAN FRAMEWORK FOR STABILITY ANALYSIS OF GNNS

**Challenges in Applying DMD Metrics to GNN Stability Analysis.** When adopting the DMD metric for the stability analysis of GNN models, a natural approach is to use graph-based manifolds (details in Appendix A.1) for calculating DMDs. However, directly using the input graph structure as the input graph-based manifold may not produce satisfactory results as shown in our empirical results presented in Table 2 and Appendix A.8. This inadequacy stems from the fact that the original input graph data (including node features) may not reside near a low-dimensional manifold (Bruna et al., 2013), while meaningful DMD-based stability analysis requires both the input and output data samples to lie near low-dimensional manifolds (Cheng et al., 2021). Therefore, a naive application of DMD metrics on the original graph structure is insufficient for assessing the stability of GNNs.

### 3.1 OVERVIEW OF SAGMAN

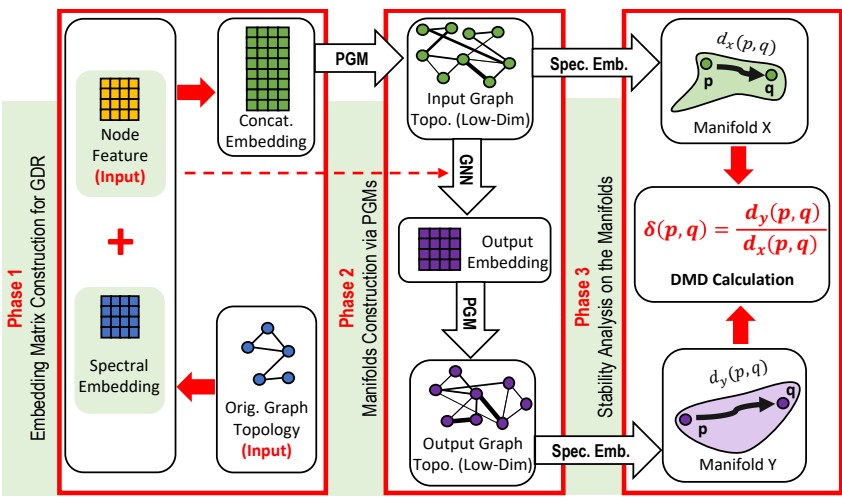

Figure 1: The proposed SAGMAN framework for stability analysis of GNNs on the manifolds.

To extend the applicability of the DMD metric to GNN settings, we introduce SAGMAN, a spectral framework for stability analysis of GNNs. A key component of SAGMAN is a novel distance-preserving Graph Dimensionality Reduction (GDR) algorithm that transforms the original input graph data—including node features and graph topology, which may reside in high-dimensional space—into a low-dimensional graph-based manifold.

As illustrated in Figure 1, the SAGMAN framework comprises three main phases:

- **Phase 1**: Creation of the input graph embedding matrix based on both node features and spectral properties of the graph. This embedding matrix is essential for the subsequent GDR step.

- **Phase 2**: Construction of low-dimensional input and output graph-based manifolds using a Probabilistic Graphical Model (PGM) approach.

- **Phase 3**: Node stability evaluation using the DMD metric, leveraging a spectral graph embedding scheme that utilizes generalized Laplacian eigenvalues and eigenvectors.

Detailed descriptions of each phase are provided in the following sections. The complete algorithmic flow of SAGMAN is shown in Section 3.5.

## 3.2 Phase 1: Embedding Matrix Construction for GDR

The calculation of the DMD metric inherently relies on pairwise distances between nodes. For graph-based manifolds, two widely used metrics for evaluating pairwise node distances are: **(1)** the shortest-path distance and **(2)** the effective resistance distance. As discussed in Appendix A.1, the effective resistance distance is more closely related to the graph's structural (spectral) properties, providing a more meaningful measure of connectivity between nodes. Moreover, previous studies have demonstrated a significant correlation between effective resistance distances and the stability of machine learning models (Cheng et al., 2021). Therefore, our focus is on performing dimensionality reduction on the input graph while preserving its original effective resistance distances.

**Graph Dimensionality Reduction via Laplacian Eigenmaps.** To achieve dimensionality reduction of the input graph, we utilize the widely recognized nonlinear dimensionality reduction algorithm, *Laplacian Eigenmaps* (Belkin & Niyogi, 2003). The Laplacian Eigenmaps algorithm begins by constructing an undirected graph where each node represents a high-dimensional data sample, and edges encode the similarities between data samples. It then computes the eigenvectors corresponding to the smallest eigenvalues of the graph Laplacian matrix, which are used to map each node into a low-dimensional space while preserving the local relationships between the data samples.

**Spectral Embedding with Eigengaps.** For a graph with $N$ nodes, a straightforward approach to apply Laplacian Eigenmaps is to compute an $N \times N$ spectral embedding matrix using the complete set of graph Laplacian eigenvectors and eigenvalues (Ng et al., 2001), representing each node with an $N$-dimensional vector. However, computing the full set of eigenvalues and eigenvectors is computationally prohibitive for large graphs.

To address this issue, we leverage a theoretical result from spectral graph clustering (Peng et al., 2015), which shows that the existence of a significant gap between consecutive eigenvalues—known as an *eigengap*—implies that the graph can be well represented in a lower-dimensional space. Specifically, the eigengap is defined as $\Upsilon(k) = \frac{\lambda_{k+1}}{\rho(k)}$, where $\rho(k)$ denotes the $k$-way expansion constant, and $\lambda_{k+1}$ is the $(k+1)$-th smallest eigenvalue of the normalized Laplacian matrix. A significant eigengap indicates the existence of a $k$-way partition where each cluster has low conductance, meaning the graph is well-clustered. Based on this, we can approximate the spectral embedding using only the first $k$ eigenvalues and eigenvectors. We define the weighted spectral embedding matrix as follows:

**Definition 3.1.** For a connected graph $G = (V, E, w)$ with its $k$ smallest nonzero Laplacian eigenvalues denoted by $0 < \lambda_1 \leq \lambda_2 \leq \ldots \leq \lambda_k$ and corresponding eigenvectors $u_1, u_2, \ldots, u_k$, the weighted spectral embedding matrix is defined as $U_k = \left[\frac{u_1}{\sqrt{\lambda_1}}, \ldots, \frac{u_k}{\sqrt{\lambda_k}}\right] \in \mathbb{R}^{|V| \times k}$.

For a graph with a significant eigengap $\Upsilon(k)$, this embedding matrix allows us to represent each node with a $k$-dimensional vector such that the effective resistance distance between any pair of nodes can be well approximated by $d^{\text{eff}}(p, q) \approx \|U_k^\top e_{p,q}\|_2^2$, where $e_p \in \mathbb{R}^{|V|}$ is the standard basis vector with a 1 at position $p$ and zeros elsewhere, and $e_{p,q} = e_p - e_q$.

**Using Eigengaps for Graph Dimension Estimation.** Determining the precise graph dimension required to embed a graph into Euclidean space while preserving certain properties (e.g., unit edge lengths) is an NP-hard problem (Erdös et al., 1965; Schaefer, 2012). However, the presence of a significant eigengap $\Upsilon(k)$ suggests that the graph can be well represented in a $k$-dimensional space (Peng et al., 2015), making $k$ an approximate measure of the graph's intrinsic dimensionality. While computing the exact value of $\Upsilon(k)$ may be challenging in practice, we can use the identified eigengap as an indicator of the suitability of SAGMAN for a given graph: graphs with significant eigengaps are more suitable to our framework since they can be effectively represented in low-dimensional spaces. Empirically, for datasets with $c$ classes, we can approximate $k$ as $k \approx 10c$ to effectively capture significant eigengaps (Deng et al., 2022).

## 3.3 Phase 2: Manifold Construction via PGMs

The embedding matrix $U_k$, derived in Phase 1 as defined in Definition 3.1, serves as the foundation for graph-based manifold construction in Phase 2. While the original Laplacian Eigenmaps algorithm suggests constructing graph-based manifolds using $k$-nearest-neighbor graphs, we find that these

manifolds do not adequately preserve the original effective resistance distances. Consequently, they are not suitable for GNN stability analysis, as empirically demonstrated in Appendix A.6.

**Probabilistic Graphical Models (PGMs) for Graph-based Manifold Learning.** PGMs, also known as Markov Random Fields (MRFs), are powerful tools in machine learning and statistical physics for representing complex systems with intricate dependency structures (Roy et al., 2009). PGMs encode the conditional dependencies between random variables through an undirected graph structure (see Appendix A.3 for more details). Recent studies have shown that the graph structure learned through PGMs can have resistance distances that encode the Euclidean distances between their corresponding data samples (Feng, 2021).

In our context, each column vector in the embedding matrix $U_k$ (as defined in Definition 3.1) corresponds to a data sample used for graph topology learning. By constructing the low-dimensional graph-based manifold via PGMs, we can effectively preserve the resistance distances from the original graph. Empirical evidence, detailed in Appendix A.6, supports that preserving these distances is essential for distinguishing between stable and unstable nodes. Therefore, our methodology leverages PGMs to maintain accurate effective resistance distances.

However, existing methods for learning PGMs may require numerous iterations to achieve convergence (Feng, 2021), limiting their applicability to large graphs.

**Scalable PGM via Spectral Sparsification.** In the proposed SAGMAN framework, we employ PGMs to create low-dimensional input graph-based manifold $G_X = (V, E_X)$ using the embedding matrix $U_k$ from Definition 3.1, and output manifold $G_Y = (V, E_Y)$ using the GNN's post-*softmax* vectors, as illustrated in Figure 1. Below, we detail the construction of the input manifold; the output manifold can be constructed similarly. Given the input embedding matrix $X = U_k \in \mathbb{R}^{|V| \times k}$, the maximum likelihood estimation (MLE) of the precision matrix $\Theta$ (PGM) can be obtained by solving the following convex optimization problem (see Appendix A.3 for more details) (Dong et al., 2019):

$$\max_{\Theta} \quad F(\Theta) = \log \det(\Theta) - \frac{1}{k} \operatorname{Tr}(X^\top \Theta X), \tag{2}$$

where $\Theta = \mathcal{L} + \frac{1}{\sigma^2} I$, $\operatorname{Tr}(\cdot)$ denotes the trace of a matrix, $\mathcal{L}$ is a valid Laplacian matrix, $I$ is the identity matrix, and $\sigma^2 > 0$ is a prior feature variance. To solve this, we give the following theorem:

**Theorem 3.2.** *Maximizing the objective function in Equation 2 can be achieved in nearly-linear time via the following edge pruning strategy equivalent to spectral sparsification of the initial dense nearest-neighbor graph. Specifically, edges with small distance ratios*

$$\rho_{p,q} = \frac{d^{\mathrm{eff}}(p,q)}{d^{\mathrm{dat}}(p,q)} = w_{p,q} \, d^{\mathrm{eff}}(p,q)$$

*are pruned, where $d^{\mathrm{eff}}(p,q)$ is the effective resistance distance between nodes $p$ and $q$, $d^{\mathrm{dat}}(p,q) = \|X_p - X_q\|_2^2$ is the data distance between the embeddings of nodes $p$ and $q$, and $w_{p,q} = \dfrac{1}{d^{\mathrm{dat}}(p,q)}$ is the weight of edge $(p,q)$.*

The proof for Theorem 3.2 is available in Appendix A.4.

**Spectral Sparsification via Graph Decomposition.** Computing the edge sampling probability $\rho_{p,q}$ for each edge $(p,q)$ requires solving Laplacian matrices multiple times (Spielman & Srivastava, 2008), making the original sparsification method computationally expensive for large graphs. An alternative approach employs a short-cycle graph decomposition scheme (Chu et al., 2020), which partitions an unweighted graph $G$ into multiple disjoint cycles by removing a fixed number of edges while ensuring a bound on the length of each cycle. However, such methods are limited to unweighted graphs.

**Lemma 3.3.** *Spectral sparsification of an undirected graph $G$, with Laplacian $L_G$, can be achieved by leveraging a short-cycle decomposition algorithm that returns a sparsified graph $H$, with Laplacian $L_H$, such that for all real vectors $x$, $x^\top L_G x \approx x^\top L_H x$ (Chu et al., 2020).*

To extend these methods to weighted graphs, we introduce an improved spectral sparsification algorithm, illustrated in Figure 2. Our approach utilizes a *low-resistance-diameter* (LRD) decomposition scheme to limit the length of each cycle as measured by the effective resistance metric. This method is particularly effective for sparsifying weighted graphs.

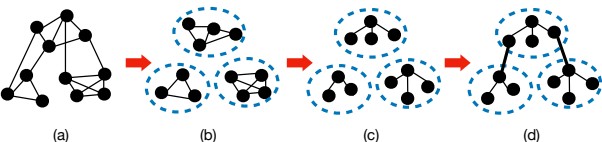

Figure 2: The proposed spectral sparsification algorithm. (a) The initial graph. (b) LRD decomposition for graph clustering. (c) LSSTs for pruning non-critical edges within clusters. (d) The final graph-based manifold with two inter-cluster edges.

The key idea is to efficiently compute the effective resistance of each edge (see Appendix A.9 for details) and employ a multilevel framework to decompose the graph into several disjoint clusters bounded by an effective resistance threshold. Importantly, the inter-cluster edges identified during this process can be inserted back into the original graph to significantly enhance the stability of GNN models, as demonstrated in Section 4.5.

### 3.4 PHASE 3: STABILITY ANALYSIS ON THE MANIFOLDS

In this phase, we analyze the stability of GNNs by quantifying the distortions between the input and output graph-based manifolds constructed in Phase 2, utilizing the Distance Mapping Distortion (DMD) metric.

**DMD with Effective Resistance Distance Metric.** Let $M$ be the mapping function of a machine learning model that transforms input data $X$ into output data $Y$, i.e., $Y = M(X)$. To assess the stability of $M$, we employ the DMD metric, which measures how distances between data samples are distorted by $M$. On the input and output graph-based manifolds, we use the *effective resistance distance* as the metric between nodes. The effective resistance distance between nodes $p$ and $q$ on the input manifold $G_X$ is computed as $d_X^{\text{eff}}(p,q) = e_{p,q}^\top L_X^+ e_{p,q}$, and similarly on the output manifold $G_Y$ as $d_Y^{\text{eff}}(p,q) = e_{p,q}^\top L_Y^+ e_{p,q}$, where $L_X^+$ and $L_Y^+$ denote the Moore–Penrose pseudoinverses of the Laplacian matrices $L_X$ and $L_Y$ of the input and output manifolds, respectively, and $e_{p,q} = e_p - e_q$ with $e_p$ being the standard basis vector corresponding to node $p$. The DMD between nodes $p$ and $q$ is defined as the ratio of the output distance to the input distance:

$$\delta^M(p,q) = \frac{d_Y^{\text{eff}}(p,q)}{d_X^{\text{eff}}(p,q)} = \frac{e_{p,q}^\top L_Y^+ e_{p,q}}{e_{p,q}^\top L_X^+ e_{p,q}}. \tag{3}$$

To quantify the maximum distortion introduced by $M$, we consider the maximum DMD over all pairs of distinct nodes: $\delta_{\max}^M$. According to Lemma A.2 in (Cheng et al., 2021), the optimal Lipschitz constant $K^*$ of the mapping $M$ is bounded by the largest eigenvalue of $L_Y^+ L_X$ and the maximum DMD:

$$\delta_{\max}^M \le K^* \le \lambda_{\max}(L_Y^+ L_X). \tag{4}$$

This relationship allows us to assess the stability of $M$ using spectral properties of the Laplacian matrices.

**Node Stability Score via Spectral Embedding.** Building on the theoretical insights from (Cheng et al., 2021), we utilize the largest generalized eigenvalues and their corresponding eigenvectors of $L_Y^+ L_X$ to evaluate the stability of individual nodes in GNNs. We compute the weighted eigensubspace matrix $V_s \in \mathbb{R}^{|V| \times s}$ for spectral embedding of the input manifold $G_X = (V, E_X)$, where $|V|$ is the number of nodes. The matrix $V_s$ is defined as: $V_s = \left[ v_1 \sqrt{\zeta_1}, v_2 \sqrt{\zeta_2}, \ldots, v_s \sqrt{\zeta_s} \right]$, where $\zeta_1 \ge \zeta_2 \ge \cdots \ge \zeta_s$ are the largest $s$ eigenvalues of $L_Y^+ L_X$, and $v_1, v_2, \ldots, v_s$ are the corresponding eigenvectors. Using $V_s$, we embed the nodes of $G_X$ into an $s$-dimensional space by representing each node $p$ with the $p$-th row of $V_s$. The stability of an edge $(p,q) \in E_X$ can then be estimated by

computing the spectral embedding distance between nodes $p$ and $q$: $\|V_s^\top e_{p,q}\|_2^2$. To assess the stability of individual nodes, we define the stability score of node $p$ as the average embedding distance to its neighbors in the input manifold:

$$\text{score}(p) = \frac{1}{|\mathbb{N}_X(p)|} \sum_{q \in \mathbb{N}_X(p)} \|V_s^\top e_{p,q}\|_2^2, \tag{5}$$

where $\mathbb{N}_X(p)$ denotes the set of neighbors of node $p$ in $G_X$. Since $\|V_s^\top e_{p,q}\|_2^2$ is proportional to $\left(\delta^M(p,q)\right)^3$, the node stability score effectively serves as a surrogate for the local Lipschitz constant, analogous to $\|\nabla_X M(p)\|$ under the manifold setting (Cheng et al., 2021). For a more detailed derivation and theoretical justification, please refer to Appendix A.11.

## 3.5 ALGORITHM FLOW OF SAGMAN

The Algorithm 1 shows the key steps in SAGMAN.

---

**Algorithm 1** SAGMAN Algorithm

---

**Input:** Graph $G = (V, E)$, node features $X$, GNN model
**Output:** Stability scores for all nodes

1. **Compute Spectral Embeddings (GDR):**
   - Compute the spectral embedding matrix $U_k$ of $G$ using its weighted Laplacian.
2. **Augment Node Features:**
   - Concatenate $X$ and $U_k$ to form the feature matrix $FM = [U_k, X]$.
3. **Construct Input Graph-based Manifold $G_X$ (PGM):**
   - Build a k-NN graph $G_{\text{dense}}^X$ using $FM$.
   - Apply spectral sparsification to $G_{\text{dense}}^X$ to obtain $G_X$.
4. **Apply GNN Model:**
   - Compute output representations $Y = \text{GNN}(G_X, X)$.
5. **Construct Output Graph-based Manifold $G_Y$ (PGM):**
   - Build a k-NN graph $G_{\text{dense}}^Y$ using $Y$.
   - Apply spectral sparsification to $G_{\text{dense}}^Y$ to obtain $G_Y$.
6. **Compute Stability Scores (DMD):**
   - Compute Laplacians $L_X$ and $L_Y$ of $G_X$ and $G_Y$.
   - Solve the generalized eigenvalue problem $L_Y V_k = \lambda L_X V_k$ to obtain $V_k$.
   - For each node $p \in V$:
     - Compute the stability score: $\text{score}(p) = \frac{1}{|\mathcal{N}_X(p)|} \sum_{q \in \mathcal{N}_X(p)} \|V_k^\top (e_p - e_q)\|_2^2$ where $\mathcal{N}_X(p)$ are the neighbors of $p$ in $G_X$.

---

## 3.6 TIME COMPLEXITY OF SAGMAN

The proposed SAGMAN framework is designed to be efficient and scalable for large graphs. Below, we analyze the time complexity of its key components. We utilize fast multilevel eigensolvers to compute the first $c$ Laplacian eigenvectors. These eigensolvers operate in nearly linear time, $O(c|V|)$, without loss of accuracy (Zhao et al., 2021), where $|V|$ denotes the number of nodes in the graph. To construct the initial graph for manifold learning, we employ the $k$-nearest neighbor algorithm, which has a nearly linear computational complexity of $O(|V|\log|V|)$ (Malkov & Yashunin, 2018). The spectral sparsification step, performed via Low-Resistance-Diameter (LRD) decomposition, has a time complexity of $O(|V|dm)$, where $d$ is the average degree of the graph, and $m$ is the order of the Krylov subspace used in the computation. By leveraging fast generalized eigensolvers (Koutis et al., 2010; Cucuringu et al., 2016), we can compute all DMD values in $O(|E|)$ time, where $|E|$ denotes the number of edges in the graph. Importantly, we ignore $d$ (average degree) because most real-world graphs are sparse (Miao et al., 2019), with average degrees much smaller than the number of nodes. In such cases, $d$ can be treated as a constant or as growing slowly with $|V|$.

Overall, the dominant terms in the time complexity are nearly linear with respect to the size of the graph. This near-linear scalability allows SAGMAN to handle large graph datasets efficiently. Results for runtime scalability are available in Appendix A.13

# 4 EXPERIMENTS

We validate our proposed SAGMAN framework through comprehensive experiments. First, we compare exact resistance distances from the original graph with their approximations derived from the constructed manifold to demonstrate the manifold's fidelity. Next, we present numerical experiments showcasing the effectiveness of our metric in quantifying GNN stability. We then highlight the efficacy of our SAGMAN-guided approach in executing graph adversarial attacks. Finally, we demonstrate how leveraging the low-dimensional input manifold created by SAGMAN significantly enhances GNN stability. Details of our experimental setup are provided in Appendix A.5. We define a node as unstable if small perturbations in the input lead to significant changes in the output, indicated by a high DMD value. Conversely, a stable node has a low DMD value, reflecting robustness to input variations.

## 4.1 EVALUATION OF GRAPH DIMENSION REDUCTION (GDR)

A natural concern arises regarding how much the constructed input graph-based manifold might change the original graph's structure. To this end, we compare the exact resistance distances calculated using the complete set of eigenpairs with approximate resistance distances estimated using the embedding matrix $U_k$ (as defined in Definition 3.1), considering various values of $k$ as shown in Table 1. Our empirical results demonstrate that using a small number of eigenpairs can effectively approximate the original effective-resistance distances. Furthermore, Table 2 shows that the SAGMAN-guided stability analysis with GDR consistently distinguishes between stable and unstable nodes, whereas the analysis without GDR fails to do so. Additional related results for various datasets and GNN architectures are provided in Appendix A.8.

Table 1: Resistance-distance preservation for the Cora graph, evaluating 100 randomly selected node pairs. Larger correlation coefficients (CC) indicate more accurate estimations.

| $k$ | 20 | 30 | 50 | 100 | 200 | 400 | 500 |
|---|---|---|---|---|---|---|---|
| CC | 0.69 | 0.78 | 0.82 | 0.87 | 0.93 | 0.97 | 0.99 |

Table 2: Cosine similarities between original and perturbed node embeddings for stable/unstable nodes under Nettack adversarial attacks on the Cora dataset using GCN. Higher cosine similarities for stable nodes and lower for unstable nodes indicate better distinction. Better results are in **bold**.

| Nettack Level | 1 | 2 | 3 |
|---|---|---|---|
| w/o GDR | 0.90/0.96 | 0.84/0.93 | 0.81/0.91 |
| w/ GDR | **0.99/0.90** | **0.98/0.84** | **0.97/0.81** |

Table 3: Comparison of Nettack/FGA error rates for 40 nodes selected using Nettack's recommendation, confidence ranking, and SAGMAN. All nodes chosen by SAGMAN-guided methods are correctly classified before perturbation. Better results are highlighted in **bold**.

| Selection | Nettack's default | Confidence Ranking | SAGMAN |
|---|---|---|---|
| Cora | 0.725/0.850 | 0.925/0.775 | **0.975/0.975** |
| Cora-ml | 0.750/0.850 | 0.800/0.700 | **0.950/0.950** |
| Citeseer | 0.800/0.875 | 0.925/0.950 | **1.000/0.975** |
| Pubmed | 0.750/0.875 | 0.750/0.925 | **0.825/0.950** |

## 4.2 METRICS FOR GNN STABILITY EVALUATION

We empirically demonstrate SAGMAN's ability to distinguish between stable and unstable samples under DICE attack (Waniek et al., 2018) and Gaussian perturbation, as shown in Figure 3. Additional results on PGD attack, various datasets—including large-scale datasets—and different GNN architectures under Nettack attacks (Zügner et al., 2018) are provided in Appendix A.10, further illustrating the effectiveness of our metric. SAGMAN is most effective for GNNs that perform feature

Table 4: Error rates for the Cora and Citeseer datasets before/after Nettack and IG-attack evasion attacks, with and without SAGMAN enhancement. 'Size' indicates the fraction of the nodes selected. Better results are highlighted in **bold**.

| Nettack Attack Budget Size | | 1% | 5% | 10% | 15% | 20% |
|---|---|---|---|---|---|---|
| Cora | w/o SAGMAN | 0.00/0.79 | 0.01/0.86 | 0.01/0.90 | **0.02**/0.93 | **0.02**/0.92 |
| | w/ SAGMAN | **0.00/0.00** | **0.00/0.00** | **0.01/0.01** | 0.06/**0.06** | 0.08/**0.08** |
| Citeseer | w/o SAGMAN | **0.00**/0.76 | **0.03**/0.82 | **0.04**/0.85 | **0.04**/0.86 | **0.03**/0.85 |
| | w/ SAGMAN | 0.09/**0.42** | 0.10/**0.37** | 0.12/**0.37** | 0.11/**0.35** | 0.08/**0.28** |
| IG-attack Budget Size | | 1% | 5% | 10% | 15% | 20% |
| Citeseer | w/o SAGMAN | 0.00/0.76 | 0.02/0.74 | 0.02/0.74 | 0.02/0.72 | 0.02/0.75 |
| | w/ SAGMAN | **0.00/0.05** | **0.02/0.07** | **0.02/0.07** | **0.02/0.04** | **0.02/0.09** |
| Cora | w/o SAGMAN | 0.00/0.96 | 0.02/0.83 | 0.02/0.81 | 0.02/0.82 | 0.02/0.81 |
| | w/ SAGMAN | **0.00/0.00** | **0.02/0.02** | **0.02/0.03** | **0.02/0.02** | **0.02/0.04** |

Table 5: Baseline and Perturbed error rates for the Citeseer dataset using GOOD-AT, SAGMAN, and both. Our experiments' attack method and backbone architecture follow those from Li et al. (2024). The best results are marked in **bold**, and the second-best results are underlined.

| Edge Attack Budget | 25 | 109 | 219 | 302 | 410 | 550 |
|---|---|---|---|---|---|---|
| GCN(baseline) | 0.2968 | 0.3211 | 0.3590 | 0.3780 | 0.4064 | 0.4301 |
| GOOD-AT | 0.2808 | 0.2820 | 0.2850 | 0.2838 | 0.2855 | 0.2915 |
| SAGMAN | 0.2808 | 0.2802 | 0.2808 | 0.2820 | 0.2814 | 0.2838 |
| GOOD-AT+SAGMAN | **0.2684** | **0.2695** | **0.2684** | **0.2690** | **0.2690** | **0.2701** |

smoothing, common in homophilic graphs. However, for heterophilic graphs, we still observed SAGMAN distinguish stable and unstable nodes, as shown in Figure 10

## 4.3 STABILITY OF GNN-BASED RECOMMENDATION SYSTEMS

We evaluate the stability of GNN-based recommendation systems using the PinSage framework (Ying et al., 2018) on the MovieLens 1M dataset (Harper & Konstan, 2015). To construct the input graph, we first homogenize the various node and edge types into a unified format. We then calculate the weighted spectral embedding matrix $U_k$ (as defined in Definition 3.1) and extract user-type samples to build a low-dimensional input (user) graph-based manifold. For the output graph-based manifold, we utilize PinSage's final output, which includes the top-10 recommended items for each user. We construct this output (user) graph-based manifold based on Jaccard similarity measures of these recommendations. Table 6 demonstrates the effectiveness of SAGMAN in distinguishing between stable and unstable users.

Table 6: Comparison of the mean Jaccard similarity (MJS) between SAGMAN-selected stable/unstable users at a perturbation level ($l$) ranging from 1 to 5 where each selected user is connected to $l$ randomly chosen new items. The MJS is computed over 20 iterations for each perturbation.

| Perturbation Level | 1 | 2 | 3 | 4 | 5 |
|---|---|---|---|---|---|
| Stable Users | 0.8513 | 0.8837 | 0.8295 | 0.8480 | 0.8473 |
| Unstable Users | 0.7885 | 0.7955 | 0.8167 | 0.8278 | 0.7794 |

## 4.4 SAGMAN-GUIDED ADVERSARIAL TARGETED ATTACK

We adapt our GNN training methodology from previous work (Jin et al., 2021). Using GCN as our base model for the Citeseer, Cora, Cora-ML, and Pubmed datasets, we employ Nettack and FGA (Chen et al., 2018) as benchmark attack methods. For target node selection, we compare Net-

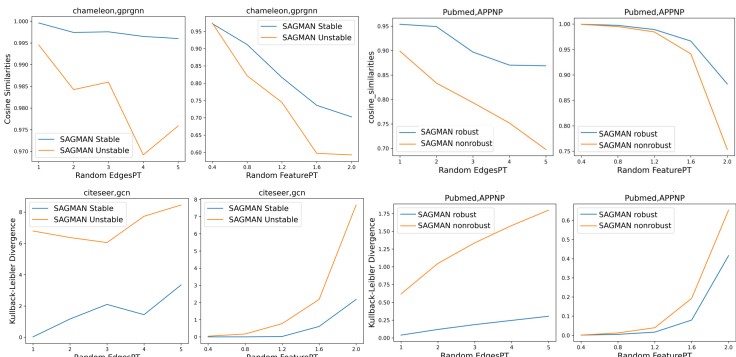

Figure 3: The horizontal axes, denoted by $X$, represent the perturbation applied. 'Random EdgesPT' refers to the DICE attack. 'Random FeaturePT' indicates the application of Gaussian noise perturbation, expressed as $X\eta$ perturbation, where $\eta$ represents Gaussian noise. SAGMAN Stable/Unstable denotes the samples classified as stable or unstable by SAGMAN, respectively.

tack's recommendation (Zügner et al., 2018), SAGMAN-guided strategies, and a heuristic confidence ranking (Chang et al., 2017).

Table 3 presents the error rates after applying Nettack and FGA attacks. The results demonstrate that SAGMAN-guided attacks outperform both Nettack's recommendation and the confidence ranking, leading to more effective adversarial attacks.

### 4.5 SAGMAN-GUIDED GNN STABILITY ENHANCEMENT

A naive approach to improving the stability of a GNN model is to replace the entire input graph with the low-dimensional graph-based manifold for GNN predictions. However, due to the significantly increased densities in the graph-based manifold, the GNN prediction accuracy may be adversely affected. To achieve a flexible trade-off between model stability and prediction accuracy, we select only the inter-cluster edges from the input graph-based manifold, as shown in Figure 2(d), and insert them into the original graph. Since the inter-cluster (bridge-like) edges are typically spectrally critical—with high sampling probabilities as defined in Theorem 3.2—adding them to the original graph significantly alters its structural (spectral) properties.

Whereas previous work (Deng et al., 2022) focuses on improving the robustness of GNNs against poisoning attacks, our work centers on enhancing robustness against evasion attacks. In Table 4, we present the error rates for both the original and the enhanced graphs, where Nettack was employed to perturb SAGMAN-selected most unstable samples within each graph. Table 5 shows the error rates for the state-of-the-art GOOD-AT (Li et al., 2024) method and our proposed SAGMAN approach. Since SAGMAN is a versatile plug-in method that can be combined with other robustness techniques, we also report results for the combined application of GOOD-AT and SAGMAN. Notably, reintegrating selected edges into the original graph significantly reduces the error rate when subjected to adversarial evasion attacks. To further demonstrate SAGMAN's effectiveness under poisoning attacks and in comparison with Lipschitz-based methods, we present additional results in Appendix A.13.

## 5 CONCLUSIONS

In this work, we introduced SAGMAN, a novel framework for analyzing the stability of GNNs at the individual node level by assessing the resistance-distance distortions between low-dimensional input and output graph-based manifolds. A key component of SAGMAN is the proposed Graph Dimensionality Reduction (GDR) approach for constructing resistance-preserving manifolds, which enables effective stability analysis.

Our experimental results demonstrate that SAGMAN effectively quantifies GNN stability, leading to significantly enhanced targeted adversarial attacks and improved GNN robustness. The current SAGMAN framework is particularly effective for graphs that can be well represented in low-dimensional spaces and exhibit large eigengaps.

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

## A APPENDIX

### A.1 GRAPH-BASED MANIFOLDS AND DISTANCE METRICS

**Graph-based Manifolds.** A manifold is a topological space that locally resembles Euclidean space near each point (Lee, 2012). In our work, we utilize *graph-based manifolds* to represent complex data structures. Specifically, we represent each manifold as an undirected (connected) graph $G = (V, E)$, where $V$ is the set of vertices corresponding to data points, and $E$ is the set of edges encoding relationships (e.g., conditional dependencies) between these points (Tenenbaum et al., 2000). This representation is particularly effective when the underlying data structure can be approximated by a network of discrete points, as is common in spectral clustering and manifold learning (Belkin & Niyogi, 2003).

**Mappings Between Manifolds.** Understanding the *mappings between manifolds* is essential for transferring and comparing information across different data representations. In our context, these mappings correspond to transformations between the input and output graph-based manifolds of an ML model. Formally, a mapping $\varphi : M_X \rightarrow M_Y$ from the input manifold $M_X$ to the output manifold $M_Y$ allows us to analyze how the model transforms data points and their relationships (Pan & Yang, 2009). This is crucial for assessing the model's stability with respect to small perturbations in the input.

**Distance Metrics on Manifolds.** Measuring dissimilarities between points on manifolds requires appropriate distance metrics. While *geodesic distances*, representing the shortest paths between points on a manifold, are commonly used, they can be computationally intensive for large datasets and may not capture global structural information due to their inherently local nature (Tenenbaum et al., 2000).

**Effective Resistance Distances on Graph-based Manifolds.** To overcome these limitations, we employ the *effective resistance distance* from electrical network theory. By modeling the undirected graph as a resistive network, the effective resistance between two nodes captures both local relationships and global structural properties of the graph. This metric effectively combines the advantages of geodesic and global distances, providing a more comprehensive measure of similarity in graph-based manifolds (Tenenbaum et al., 2000). Additionally, it is computationally efficient for large-scale graphs (Klein & Zhu, 1998), making it practical for our analysis.

To illustrate the difference between geodesic distance and effective resistance distance, consider the example in Figure 4. In all three graphs, nodes $A$ and $B$ have the same geodesic distance (i.e., the shortest path length is identical). However, their effective resistance distances $\Omega_{AB}$ vary significantly due to differences in the global structure of each graph. This example demonstrates that while geodesic distance only accounts for the shortest path, effective resistance distance incorporates the overall connectivity and network topology, making it a more informative metric for assessing similarities in graph-based manifolds.

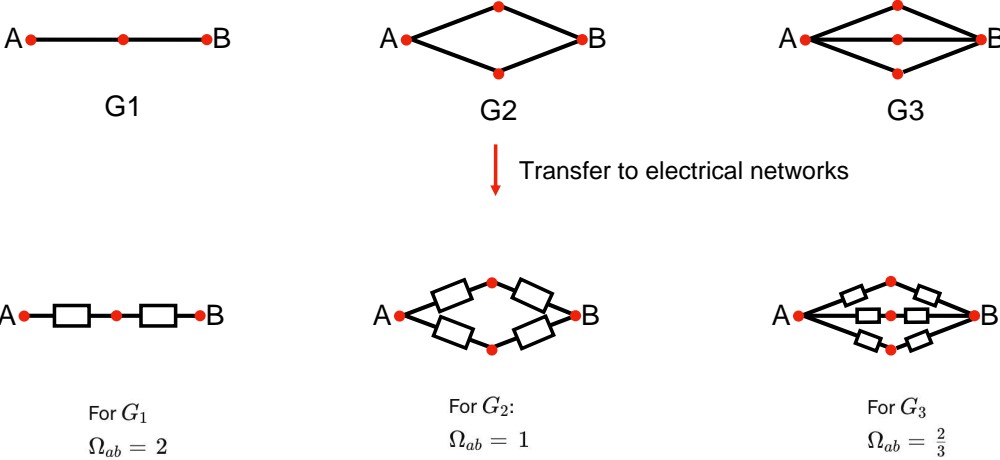

Figure 4: Examples of three different graph structures with nodes $A$ and $B$. Despite having the same geodesic distance (shortest path length), the effective resistance distances $\Omega_{AB}$ vary due to the different global structures of the graphs.

By incorporating effective resistance distances into our stability analysis, we can more accurately assess the distance distortions introduced by the mappings between the input and output manifolds (as discussed in Section 2.1). This provides a solid foundation for evaluating the stability of ML

models on graph-based manifolds, enhancing both the theoretical rigor and practical applicability of our approach.

## A.2 SPECTRAL GRAPH THEORY

Spectral graph theory is a branch of mathematics that studies the properties of graphs through the eigenvalues and eigenvectors of matrices associated with the graph (Chung, 1997). Let $G = (V, E, w)$ denote an undirected graph $G$, $V$ denote a set of nodes (vertices), $E$ denote a set of edges and $w$ denote the corresponding edge weights. The adjacency matrix can be defined as:

$$A(i, j) = \begin{cases} w(i, j) & \text{if } (i, j) \in E \\ 0 & \text{otherwise} \end{cases} \tag{6}$$

The Laplacian matrix of $G$ can be constructed by $L = D - A$, where $D$ denotes the degree matrix.

**Lemma A.1.** *(Courant-Fischer Minimax Theorem) The $k$-th largest eigenvalue of the Laplacian matrix $L \in \mathbb{R}^{|V| \times |V|}$ can be computed as follows:*

$$\lambda_k(L) = \min_{dim(U)=k} \max_{\substack{u_k \in U \\ u_k \neq 0}} \frac{u_k^\top L u_k}{u_k^\top u_k} \tag{7}$$

Lemma A.1 is the Courant-Fischer Minimax Theorem (Golub & Van Loan, 2013) for solving the eigenvalue problem: $Lu_k = \lambda_k u_k$. The generalized Courant-Fischer Minimax Theorem for solving generalized eigenvalue problem $L_X v_k = \lambda_k L_Y v_k$ can be expressed as follows:

**Lemma A.2.** *(The Generalized Courant-Fischer Minimax Theorem) Given two Laplacian matrices $L_X, L_Y \in \mathbb{R}^{|V| \times |V|}$ such that $null(L_Y) \subseteq null(L_X)$, $L_Y^+$ denotes the Moore–Penrose pseudoinverse of $L_Y$, the $k$-th largest eigenvalue of $L_Y^+ L_X$ can be computed under the condition of $1 \leq k \leq rank(L_Y)$ by:*

$$\lambda_k(L_Y^+ L_X) = \min_{\substack{dim(U)=k \\ U \perp null(L_Y)}} \max_{v_k \in U} \frac{v_k^\top L_X v_k}{v_k^\top L_Y v_k}. \tag{8}$$

## A.3 GRAPH-BASED MANIFOLD LEARNING VIA PGMs

Given $M$ samples of $N$-dimensional vectors stored in a data matrix $X \in \mathbb{R}^{N \times M}$, the recent graph topology learning methods (Kalofolias & Perraudin, 2019; Dong et al., 2019) estimate graph Laplacians from $X$ for achieving the following desired characteristics:

**Smoothness of Graph Signals.** The graph signals corresponding to the real-world data should be sufficiently smooth on the learned graph structure: the signal values will only change gradually across connected neighboring nodes. The smoothness of a signal $x$ over an undirected graph $G = (V, E, w)$ can be measured with the following Laplacian quadratic form: $x^\top L x = \sum_{(p,q) \in E} w_{p,q}(x(p) - x(q))^2$, where $w_{p,q}$ denotes the weight of edge $(p, q)$, $L = D - W$ denotes the Laplacian, $D$ denotes the diagonal (degree) matrix, and $W$ denotes the adjacency matrix of $G$, respectively. The smaller quadratic form implies the smoother signals across the edges in the graph. The smoothness ($Q$) of a set of signals $X$ over graph $G$ is computed using the following matrix trace (Dong et al., 2019): $Q(X, L) = Tr(X^\top L X)$, where $Tr(\bullet)$ denotes the matrix trace.

**Sparsity of the Estimated Graph.** Graph sparsity is another critical consideration in graph learning. One of the most important motivations of learning a graph is to use it for downstream computing tasks. Therefore, more desired graph topology learning algorithms should allow better capturing and understanding the global structure (manifold) of the data set, while producing sufficiently sparse graphs that can be easily stored and efficiently manipulated in the downstream algorithms, such as circuit simulations, network partitioning, dimensionality reduction, data visualization, etc.

**Problem Formulation.** Consider a random vector $x \sim N(0, \Sigma)$ with probability density function:

$$f(x) = \frac{\exp\left(-\frac{1}{2} x^\top \Sigma^{-1} x\right)}{(2\pi)^{N/2} \det(\Sigma)^{(1/2)}} \propto \det(\Theta)^{1/2} \exp\left(-\frac{1}{2} x^\top \Theta x\right), \tag{9}$$

where $\Sigma = \mathbb{E}[xx^\top] \succ 0$ denotes the covariance matrix, and $\Theta = \Sigma^{-1}$ denotes the precision matrix (inverse covariance matrix). Prior graph topology learning methods aim at estimating sparse precision matrix $\Theta$ from potentially high-dimensional input data, which fall into the following two categories:

**(A) The graphical Lasso** method aims at estimating a sparse precision matrix $\Theta$ using convex optimization to maximize the log-likelihood of $f(x)$ (Friedman et al., 2008):

$$\max_{\Theta} : \log \det(\Theta) - Tr(\Theta S) - \beta \|\Theta\|_1, \tag{10}$$

where $\Theta$ denotes a non-negative definite precision matrix, $S$ denotes a sample covariance matrix, and $\beta$ denotes a regularization parameter. The first two terms together can be interpreted as the log-likelihood under a Gaussian Markov Random Field (GMRF). $\|\bullet\|$ denotes the entry-wise $\ell_1$ norm, so $\beta\|\Theta\|_1$ becomes the sparsity promoting regularization term. This model learns the graph structure by maximizing the penalized log-likelihood. When the sample covariance matrix $S$ is obtained from $M$ i.i.d (independent and identically distributed) samples $X = [x_1, ..., x_M]$ where $X \sim N(0, S)$ has an $N$-dimensional Gaussian distribution with zero mean, each element in the precision matrix $\Theta_{i,j}$ encodes the conditional dependence between variables $X_i$ and $X_j$. For example, $\Theta_{i,j} = 0$ implies that variables $X_i$ and $X_j$ are conditionally independent, given the rest.

**(B) The Laplacian estimation** methods have been recently introduced for more efficiently solving the following convex problem (Dong et al., 2019; Lake & Tenenbaum, 2010):

$$\max_{\Theta} : F(\Theta) = \log \det(\Theta) - \frac{1}{M} Tr(X^\top \Theta X) - \beta \|\Theta\|_1, \tag{11}$$

where $\Theta = L + \frac{1}{\sigma^2} I$, $L$ denotes the set of valid graph Laplacian matrices, $I$ denotes the identity matrix, and $\sigma^2 > 0$ denotes prior feature variance. It can be shown that the three terms in (11) are corresponding to $\log \det(\Theta)$, $Tr(\Theta S)$ and $\beta\|\Theta\|_1$ in (10), respectively. Note that the second term also promotes graph sparsity, so the $\beta\|\Theta\|_1$ can be dropped without impacting the final solution. Since $\Theta = L + \frac{1}{\sigma^2} I$ correspond to symmetric and positive definite (PSD) matrices (or M matrices) with non-positive off-diagonal entries, this formulation will lead to the estimation of attractive GMRFs (Dong et al., 2019; Slawski & Hein, 2015). **In case $X$ is non-Gaussian**, formulation (11) can be understood as Laplacian estimation based on minimizing the Bregman divergence between positive definite matrices induced by the function $\Theta \mapsto -\log \det(\Theta)$ (Slawski & Hein, 2015).

### A.4 Proof for Theorem 3.2

In the SAGMAN framework, we aim to construct low-dimensional manifolds for both the input and output of a GNN. For the input manifold, we use the embedding matrix $X = U_k \in \mathbb{R}^{|V| \times k}$ (as defined in Definition 3.1), where $|V|$ is the number of nodes in the graph and $k$ is the dimensionality of the embedding space. The goal is to learn a precision matrix $\Theta$ that captures the underlying graph structure reflected in the embeddings.

**Maximum Likelihood Estimation (MLE) of the PGM (Precision Matrix).** We start by formulating the MLE of the precision matrix $\Theta$ as a convex optimization problem (Dong et al., 2019):

$$\max_{\Theta} \quad F(\Theta) = \log \det(\Theta) - \frac{1}{k} \text{Tr}(X^\top \Theta X), \tag{12}$$

where:

- $\Theta = \mathcal{L} + \frac{1}{\sigma^2} I$,
- $\mathcal{L}$ is the graph Laplacian matrix,
- $I$ is the identity matrix,
- $\sigma^2 > 0$ is a prior variance term,
- $\text{Tr}(\cdot)$ denotes the trace of a matrix.

**Expanding the Laplacian Matrix.** The graph Laplacian matrix $\mathcal{L}$ can be decomposed as:

$$\mathcal{L} = \sum_{(p,q) \in E} w_{p,q} e_{p,q} e_{p,q}^\top, \tag{13}$$

where:

- $E$ is the set of edges in the graph,
- $w_{p,q}$ is the weight of edge $(p, q)$,
- $e_{p,q} = e_p - e_q$, with $e_p$ being the standard basis vector corresponding to node $p$.

The edge weights are defined as:

$$w_{p,q} = \frac{1}{\|X^\top e_{p,q}\|_2^2} = \frac{1}{\|X_p - X_q\|_2^2}, \tag{14}$$

where $X_p$ and $X_q$ are the embeddings of nodes $p$ and $q$, respectively.

**Expressing the Objective Function.** Substituting $\Theta$ and $\mathcal{L}$ into the objective function, we can split $F(\Theta)$ into two parts:

$$F = F_1 - \frac{1}{k} F_2, \tag{15}$$

where:

1. First Term ($F_1$):

$$F_1 = \log \det(\Theta) = \sum_{i=1}^{|V|} \log \left( \lambda_i + \frac{1}{\sigma^2} \right), \tag{16}$$

   with $\lambda_i$ being the $i$-th eigenvalue of the Laplacian $\mathcal{L}$.

2. Second Term ($F_2$):

$$F_2 = \mathrm{Tr}(X^\top \Theta X) = \frac{\mathrm{Tr}(X^\top X)}{\sigma^2} + \sum_{(p,q) \in E} w_{p,q} \|X^\top e_{p,q}\|_2^2. \tag{17}$$

   The term $\frac{\mathrm{Tr}(X^\top X)}{\sigma^2}$ is constant with respect to $w_{p,q}$ and can be ignored for optimization over $w_{p,q}$.

**Computing Partial Derivatives.** To optimize $F$ with respect to the edge weights $w_{p,q}$, we compute the partial derivatives of $F_1$ and $F_2$:

**Derivative of $F_1$ with respect to edge weight:**

$$\frac{\partial F_1}{\partial w_{p,q}} = \sum_{i=1}^{|V|} \frac{1}{\lambda_i + \frac{1}{\sigma^2}} \frac{\partial \lambda_i}{\partial w_{p,q}}. \tag{18}$$

Since $\frac{\partial \lambda_i}{\partial w_{p,q}} = v_i^\top \frac{\partial \mathcal{L}}{\partial w_{p,q}} v_i$, where $v_i$ is the eigenvector corresponding to $\lambda_i$, and $\frac{\partial \mathcal{L}}{\partial w_{p,q}} = e_{p,q} e_{p,q}^\top$, we have:

$$\frac{\partial F_1}{\partial w_{p,q}} = \sum_{i=1}^{|V|} \frac{(v_i^\top e_{p,q})^2}{\lambda_i + \frac{1}{\sigma^2}}. \tag{19}$$

When $\sigma$ approaches infinity, this expression is known as the **effective resistance distance** between nodes $p$ and $q$:

$$d^{\text{eff}}(p,q) = e_{p,q}^{\top} \Theta^{-1} e_{p,q}. \tag{20}$$

So,

$$\frac{\partial F_1}{\partial w_{p,q}} = d^{\text{eff}}(p,q). \tag{21}$$

**Derivative of $F_2$ with respect to edge weight:**

$$\frac{\partial F_2}{\partial w_{p,q}} = \|X^{\top} e_{p,q}\|_2^2 = \|X_p - X_q\|_2^2. \tag{22}$$

This is the **data distance** between nodes $p$ and $q$:

$$d^{\text{dat}}(p,q) = \|X_p - X_q\|_2^2 = \frac{1}{w_{p,q}}. \tag{23}$$

Since $w_{p,q} = \frac{1}{d^{\text{dat}}(p,q)}$, we have:

$$\frac{\partial F_2}{\partial w_{p,q}} = \frac{1}{w_{p,q}}. \tag{24}$$

**Optimization Strategy.** The gradient of $F$ with respect to $w_{p,q}$ is:

$$\frac{\partial F}{\partial w_{p,q}} = \frac{\partial F_1}{\partial w_{p,q}} - \frac{1}{k}\frac{\partial F_2}{\partial w_{p,q}} = d^{\text{eff}}(p,q) - \frac{1}{k}\frac{1}{w_{p,q}}. \tag{25}$$

To maximize $F$, we can:

- **Increase** $w_{p,q}$ when $d^{\text{eff}}(p,q) > \frac{1}{k}\frac{1}{w_{p,q}}$.

- **Decrease** $w_{p,q}$ when $d^{\text{eff}}(p,q) < \frac{1}{k}\frac{1}{w_{p,q}}$.

However, since $w_{p,q} \geq 0$, decreasing $w_{p,q}$ effectively means pruning the edge $(p,q)$.

**Edge Pruning Strategy.** We aim to prune edges where:

- **Effective Resistance Distance is Small** ($d^{\text{eff}}(p,q)$ **is small**): The nodes are well-connected in terms of the graph structure.
- **Data Distance is Large** ($d^{\text{dat}}(p,q)$ **is large**): The embeddings of the nodes are far apart.

This leads us to consider the **distance ratio** $\rho_{p,q}$:

$$\rho_{p,q} = \frac{d^{\text{eff}}(p,q)}{d^{\text{dat}}(p,q)} = w_{p,q}\, d^{\text{eff}}(p,q). \tag{26}$$

Edges with **small** $\rho_{p,q}$ are candidates for pruning because they contribute less to maximizing $F$. By pruning such edges, we focus on retaining edges that have a significant impact on the graph's spectral properties.

**Connection to Spectral Graph Sparsification.** The ratio $\rho_{p,q}$ corresponds to the **edge sampling probability** used in spectral graph sparsification (Spielman & Teng, 2011). Spectral sparsification aims to approximate the original graph with a sparser graph that preserves its spectral (Laplacian) properties.

In spectral sparsification:

- **Edges are sampled with probability proportional to** $w_{p,q}d^{\text{eff}}(p,q)$.
- **Edges with higher** $\rho_{p,q}$ are more likely to be included in the sparsified graph.

Therefore, our edge pruning strategy—performing spectral sparsification on the initial graph—is equivalent to maximize the objective function in Equation 2. This ensures that the essential structural properties of the graph are maintained while reducing complexity.

### A.5 EXPERIMENTAL SETUP

See Appendix A.10 for detailed descriptions of all graph datasets used in this work. We employ the most popular backbone GNN models including GCN (Kipf & Welling, 2016), GPRGNN (Chien et al., 2020), GAT (Veličković et al., 2017), APPNP (Gasteiger et al., 2018), ChebNet (Defferrard et al., 2016), and Polynormer (Deng et al., 2024). The recommendation system is based on Pin-Sage (Ying et al., 2018). Perturbations include Gaussian noise evasion attacks and adversarial attacks (DICE (Waniek et al., 2018), Nettack (Zügner et al., 2018), and FGA (Chen et al., 2018)). The input graph-based manifolds are constructed using graph adjacency and node features, while the output graph-based manifold is created using post-*softmax* vectors. To showcase SAGMAN's effectiveness in differentiating stable from unstable nodes, we apply SAGMAN to select $1\%$ of the entire dataset as stable nodes, and another $1\%$ as unstable nodes. This decision stems from that only a portion of the dataset significantly impacts model stability (Cheng et al., 2021; Hua et al., 2021; Chang et al., 2017). Additional evaluation results on the entire dataset can be found in Appendix A.12. We quantify output perturbations using cosine similarity and Kullback-Leibler divergence (KLD). Additional insights on cosine similarity, KLD, and accuracy can be found in Appendix A.7. For a large-scale dataset "ogbn-arxiv", due to its higher output dimensionality, we focus exclusively on accuracy comparisons. This decision is informed by the KLD estimator's $n^{-\frac{1}{d}}$ convergence rate (Roldán & Parrondo, 2012), where $n$ is the number of samples and $d$ is the dimension. In this paper, our spectral embedding method consistently utilizes the smallest $50$ eigenpairs for all experiments, unless explicitly stated otherwise.

### A.6 MEASURE DMD WITHOUT PGMS

| Attack | Embedded Matrix Cosine Similarity (Stable / Unstable) |
|---|---|
| DICE level 1 | 0.89 / 0.89 |
| Gaussian noise 1.0 | 0.96 / 0.96 |

Table 7: The stable (unstable) nodes identified based on traditional graph-based manifolds (Belkin & Niyogi, 2003)

.

As shown in Table 7, we cannot obtain meaningful DMDs with traditional graph-based manifolds (kNN graphs) adopted in the Laplacian Eigenmaps framework (Belkin & Niyogi, 2003).

### A.7 METRICS FOR ASSESSING GNN STABILITY

In the context of single-label classification in graph nodes, consider an output vector $\mathbf{y} = [y_1, y_2, ..., y_k]$ corresponding to an input $\mathbf{x}$, where $k$ represents the total number of classes. The model's predicted class is denoted as $\hat{y} = \text{argmax}_i(y_i)$. Now, let's assume that the output vector transforms to $\mathbf{y}' = [y_1', y_2', ..., y_k']$, while preserving the ordinality of the elements, i.e., if $y_i > y_j$, then $y_i' > y_j'$. This condition ensures that $\hat{y}' = \text{argmax}_i(y_i') = \hat{y}$.

Relying solely on model accuracy can be deceptive, as it is contingent upon the preservation of the ordinality of the output vector elements, even when the vector itself undergoes significant transformations. This implies that the model's accuracy remains ostensibly unaffected as long as the ranking of the elements within the output vector is conserved. However, this perspective neglects potential alterations in the model's prediction confidence levels.

In contrast, the cosine similarity provides a more holistic measure as it quantifies the angle between two vectors, thereby indicating the extent of modification in the output direction. This method offers a more granular insight into the impact of adversarial attacks on the model's predictions.

Moreover, it is crucial to consider the nature of the output space, $Y$. In situations where $Y$ forms a probability distribution, a common occurrence in classification problems, the application of a distribution distance measure such as the Kullback-Leibler (KL) divergence is typically more suitable. Unlike the oversimplified perspective of accuracy, these measures can provide a nuanced understanding of the degree of perturbation introduced in the predicted probability distribution by an adversarial attack. This additional granularity can expose subtle modifications in the model's output that might be missed when solely relying on accuracy as a performance metric.

## A.8 GNN Stability Analysis without GDR

In this study, we present the outcomes of stability quantification using original input and output graphs, as depicted in Figure 5. Our experimental findings underscore a key observation: SAGMAN without GDR does not allow for meaningful estimations of the GNN stability.

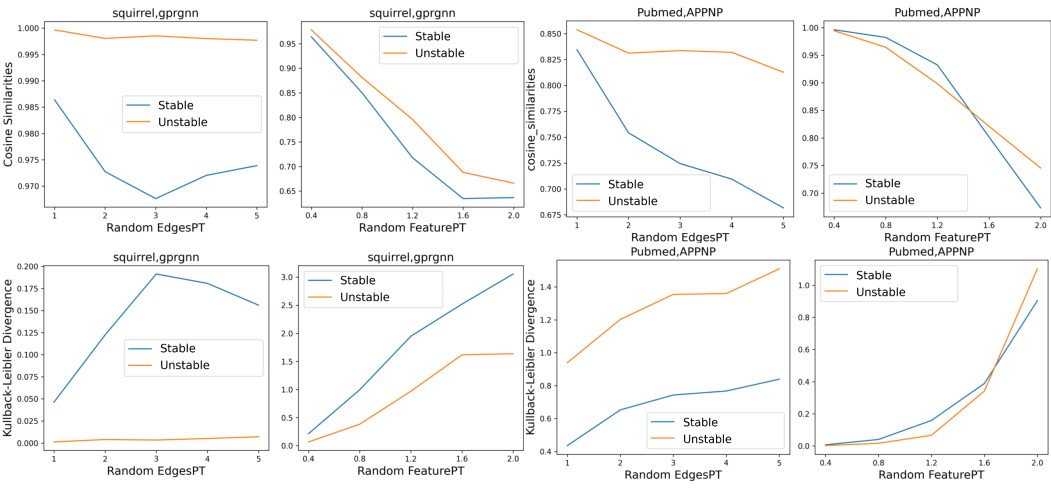

Figure 5: The horizontal axes, denoted by $X$, represent the magnitude of perturbation applied. 'Random EdgesPT' refers to the DICE adversarial attack scenario, in which pairs of nodes with different labels are connected and pairs with the same label are disconnected, with the number of pairs being equal to $X$. 'Random FeaturePT' indicates the application of Gaussian noise perturbation, expressed as $FM + X\eta$, where $FM$ denotes the feature matrix and $\eta$ represents Gaussian noise. The upper and lower subfigures illustrate the cosine similarity and the Kullback–Leibler Divergence (KLD). 'Stable/Unstable' denotes the samples that are classified as stable or unstable without GDR, respectively.

## A.9 Fast Effective-Resistance Estimation for LRD-based Graph Decomposition

The effective-resistance between nodes $(p, q) \in |V|$ can be computed using the following equation:

$$d^{eff}(p, q) = \sum_{i=2}^{N} \frac{(u_i^\top e_{p,q})^2}{u_i^\top L_G u_i}, \tag{27}$$

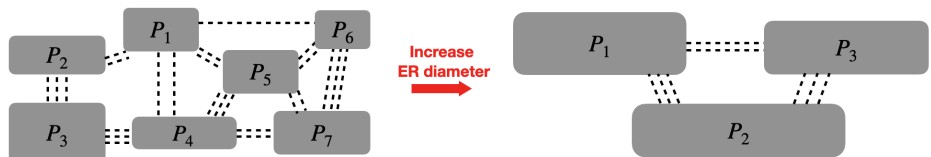

Figure 6: Graph decomposition results with respect to effective-resistance (ER) diameter

where $u_i$ represents the eigenvector corresponding to $\sigma_i$ eigenvalue of $L_G$ and $e_{p,q} = e_p - e_q$. To avoid the computational complexity associated with computing eigenvalues/eigenvectors, we leverage a scalable algorithm that approximates the eigenvectors by exploiting the Krylov subspace. In this context, given a nonsingular matrix $A_{N \times N}$ and a vector $c \neq 0 \in \mathbb{R}^N$, the order-$(m)$ Krylov subspace generated by $A$ from $c$ is defined as:

$$\kappa_m(A, c) := span(c, Ac, A^2 c, ..., A^{m-1} c), \tag{28}$$

where $c$ denotes a random vector, and $A$ denotes the adjacency matrix of graph $G$. We compute a new set of vectors denoted as $x^{(1)}, x^{(2)}, ..., x^{(m)}$ by ensuring that the Krylov subspace vectors are mutually orthogonal with unit length. We estimate the effective-resistance between node $p$ and $q$ using Equation 27 by exploring the eigenspace of $L_G$ and selecting the vectors that capture various spectral properties of $G$:

$$d^{eff}(p, q) \approx \sum_{i=1}^{m} \frac{(x^{(i)\top} e_{p,q})^2}{x^{(i)\top} L_G x^{(i)}}, \tag{29}$$

We control the diameter of each cycle by propagating effective resistances across multiple levels. Let $G = (V, E)$ represent the graph at the $\delta$-th level, and let the edge $(p, q) \in E$ be a contracted edge that creates a supernode $\vartheta \in V^{(\delta+1)}$ at level $\delta + 1$. We denote the vector of node weights as $\eta^{(\delta)} \in \mathbb{R}_{\geq 0}^{V^{(\delta)}}$, which is initially set to all zeros for the original graph. The update of $\eta$ at level $\delta + 1$ is defined as follows:

$$\eta_\vartheta := \eta(p^{(\delta)}) + \eta(q^{(\delta)}) + d_{eff}^{(\delta)}(p, q). \tag{30}$$

Consequently, the effective-resistance diameter of each cycle is influenced not only by the computed effective-resistance ($d_{eff}^{(\delta)}$) at the current level but also by the clustering information acquired from previous levels.

The graph decomposition results with respect to effective-resistance (ER) diameter are illustrated in Figure 6. The figure demonstrates that selecting a larger ER diameter leads to the decomposition of the graph into a smaller number of partitions, with more nodes included in each cluster. On the left side of the figure, the graph is decomposed into seven partitions: $P_1, ..., P_7$, by choosing a smaller ER diameter. Conversely, increasing the ER diameter on the right side of the figure results in the graph being partitioned into three clusters: $P_1, P_2,$ and $P_3$.

## A.10 ADDITIONAL RESULTS FOR GNN STABILITY EVALUATION AND STATISTICS OF DATASETS

We present the additional results in Figure 8, Figure 9, Figure 10, Figure 7, Table 8, and Table 9. Table 10 summarizes the datasets utilized.

Table 8: Robustness Evaluation of GAT under PGD Attack on Cora

| PGD Perturbation | 0.05 | 0.10 | 0.15 |
|---|---|---|---|
| Robust Accuracy | 1.0000 | 1.0000 | 1.0000 |
| Non-Robust Accuracy | 0.9630 | 0.8889 | 0.8148 |

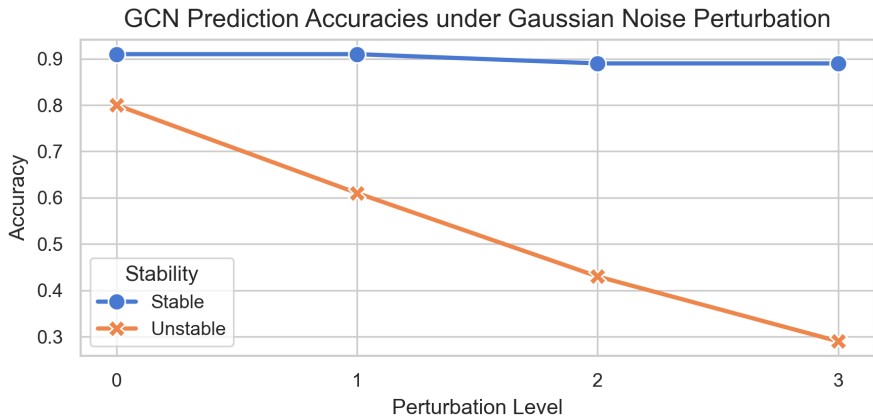

Figure 7: SAGMAN-selected stable/unstable samples for the "ogbn-arxiv" dataset. We report the GCN prediction accuracies under different levels of Gaussian Noise Perturbation $FM + X\eta$, where $FM$ denotes the feature matrix, $\eta$ represents Gaussian noise, and $X$ is the noise perturbation level.

Table 9: Nettack adversarial attack targeting selected Cora samples in GCN

| Nettack Level | Cosine Similarities: Stable/Unstable |
|---|---|
| 1 | 0.99/0.90 |
| 2 | 0.98/0.84 |
| 3 | 0.97/0.81 |

## A.11 WHY GENERALIZED EIGENPAIRS ASSOCIATE WITH DMD

(Cheng et al., 2021) propose a method to estimate the maximum distance mapping distortion (DMD), denoted as $\delta_{max}^M$, by solving the following combinatorial optimization problem:

$$\max \delta^M = \max_{\substack{\forall p,q \in V \\ p \neq q}} \frac{e_{p,q}^\top L_Y^+ e_{p,q}}{e_{p,q}^\top L_X^+ e_{p,q}} \tag{31}$$

When computing $\delta_{max}^M$ via effective-resistance distance, the stability score is an upper bound of $\delta_{max}^M$ (Cheng et al., 2021).

A function $Y = M(X)$ is called Lipschitz continuous if there exists a real constant $K \geq 0$ such that for all $x_i, x_j \in X$:

$$dist_Y(M(x_i), M(x_j)) \leq K dist_X(x_i, x_j), \tag{32}$$

where $K$ is the Lipschitz constant for the function $M$. The smallest Lipschitz constant, denoted by $K^*$, is called the best Lipschitz constant. Let the resistance distance be the distance metric, then (Cheng et al., 2021):

$$\lambda_{max}(L_Y^+ L_X) \geq K^* \geq \delta_{max}^M. \tag{33}$$

Equation 33 indicates that the $\lambda_{max}(L_Y^+ L_X)$ is also an upper bound of the best Lipschitz constant $K^*$ under the low dimensional manifold setting. A greater $\lambda_{max}(L_Y^+ L_X)$ of a function (model) implies worse stability since the output will be more sensitive to small input perturbations. A node pair $(p, q)$ is deemed non-robust if it exhibits a large DMD, i.e., $\delta^M(p, q) \approx \delta_{\max}^M$. This suggests that a non-robust node pair consists of nodes that are adjacent in the $G_X$ but distant in the $G_Y$. To effectively identify such non-robust node pairs, the Cut Mapping Distortion (CMD) metric was

Table 10: Summary of datasets used in our experiments

| Dataset | Type | Nodes | Edges | Classes | Features |
|---------|------|-------|-------|---------|----------|
| Cora | Homophily | 2,485 | 5,069 | 7 | 1,433 |
| Cora-ML | Homophily | 2,810 | 7,981 | 7 | 2,879 |
| Pubmed | Homophily | 19,717 | 44,324 | 3 | 500 |
| Citeseer | Homophily | 2,110 | 3,668 | 6 | 3,703 |
| Chameleon | Heterophily | 2,277 | 62,792 | 5 | 2,325 |
| Squirrel | Heterophily | 5,201 | 396,846 | 5 | 2,089 |
| ogbn-arxiv | Homophily | 169,343 | 1,166,243 | 40 | 128 |

introduced. For two graphs $G_X$ and $G_Y$ sharing the same node set $V$, let $S \subset V$ denote a node subset and $\bar{S}$ denote its complement. Also, let $cut_G(S, \bar{S})$ denote the number of edges crossing $S$ and $\bar{S}$ in graph $G$. The CMD $\zeta(S)$ of node subset $S$ is defined as (Cheng et al., 2021):

$$\zeta(S) \stackrel{\text{def}}{=} \frac{cut_{G_Y}(S, \bar{S})}{cut_{G_X}(S, \bar{S})}. \tag{34}$$

A small CMD score indicates that node pairs crossing the boundary of $S$ are likely to have small distances in $G_X$ but large distances in $G_Y$.

Given the Laplacian matrices $L_X$ and $L_Y$ of input and output graphs, respectively, the minimum CMD $\zeta_{\min}$ satisfies the following inequality:

$$\zeta_{\min} = \min_{\forall S \subset V} \zeta(S) \geq \frac{1}{\sigma_{\max}(L_Y^+ L_X)} \tag{35}$$

Equation 35 establishes a connection between the maximum generalized eigenvalue $\sigma_{\max}(L_Y^+ L_X)$ and $\zeta_{\min}$, indicating the ability to exploit the largest generalized eigenvalues and their corresponding eigenvectors to measure the stability of node pairs. Embedding $G_X$ with generalized eigenpairs. We first compute the weighted eigensubspace matrix $V_s \in \mathbb{R}^{N \times s}$ for spectral embedding on $G_X$ with $N$ nodes:

$$V_s \stackrel{\text{def}}{=} [v_1 \sqrt{\sigma_1}, ..., v_s \sqrt{\sigma_s}], \tag{36}$$

where $\sigma_1, \sigma_2, ..., \sigma_s$ represent the first $s$ largest eigenvalues of $L_Y^+ L_X$ and $v_1, v_2, ..., v_s$ are the corresponding eigenvectors. Consequently, the input graph $G_X$ can be embedded using $V_s$, so each node is associated with an $s$-dimensional embedding vector. We can then quantify the stability of an edge $(p, q) \in E_X$ by measuring the spectral embedding distance of its two end nodes $p$ and $q$.

Formally, we have the edge stability score defined for any edge $(p, q) \in E_X$ as $stability^M(p, q) \stackrel{\text{def}}{=} \|V_s^\top e_{p,q}\|_2^2$ Let $u_1, u_2, ..., u_s$ denote the first $s$ dominant generalized eigenvectors of $L_X L_Y^+$. If an edge $(p, q)$ is dominantly aligned with one dominant eigenvector $u_k$, where $1 \leq k \leq r$, the following holds:

$$(u_i^\top e_{p,q})^2 \approx \begin{cases} \alpha_k^2 \gg 0 & \text{if } (i = k) \\ 0 & \text{if } (i \neq k). \end{cases} \tag{37}$$

Then its edge stability score has the following connection with its DMD computed using effective-resistance distances (Cheng et al., 2021):

$$\|V_s^\top e_{p,q}\|_2^2 \propto \left(\delta^M(p, q)\right)^3. \tag{38}$$

The stability score of an edge $(p, q) \in E_X$ can be regarded as a surrogate for the directional derivative $\|\nabla_v M(x)\|$ under the manifold setting, where $v = \pm(x_p - x_q)$. An edge with a larger stability score is considered more non-robust and can be more vulnerable to attacks along the directions formed by its end nodes.

Last, the node stability score can be calculated for any node (data sample) $p \in V$ as follows:

$$score(p) = \frac{1}{|\mathbb{N}_X(p)|} \sum_{q_i \in \mathbb{N}_X(p)} \left(\|V_s^\top e_{p,q}\|_2^2\right) \propto \frac{1}{|\mathbb{N}_X(p)|} \sum_{q_i \in \mathbb{N}_X(p)} \left(\delta^M(p, q_i)\right)^3 \tag{39}$$

where $q_i \in \mathbb{N}_X(p)$ denotes the $i$-th neighbor of node $p$ in graph $G_X$, and $\mathbb{N}_X(p) \in V$ denotes the node set including all the neighbors of $p$. The DMD score of a node (data sample) $p$ can be regarded as a surrogate for the function gradient $\|\nabla_x M(p)\|$ where $x$ is near $p$ under the manifold setting. A node with a larger stability score implies it is likely more vulnerable to adversarial attacks.

## A.12 Various Sampling Schemes for SAGMAN-guided Perturbations

Previous works (Cheng et al., 2021; Hua et al., 2021; Chang et al., 2017) highlighted that only part of the dataset plays a crucial role in model stability, so we want to focus on the difference between the most "stable" and "unstable" parts. However, it is certainly feasible to evaluate the entire graph. Table 11 shows the result regarding the Pubmed dataset in GPRGNN under Gaussian noise perturbation. Samples were segmented based on SAGMAN ranking, with the bottom 20% being the most "stable", the middle 60% as intermediate, and the top 20% representing the most "unstable". As anticipated, the "stable" category (representing the bottom 20%) should exhibit the lowest average KL divergences. This is followed by the intermediate category (covering the mid 60%), and finally, the "unstable" category (comprising the top 20%) should display the highest divergences.

Table 11: KLD across varying Gaussian noise perturbations, expressed as $FM + X\eta$, where $FM$ denotes the feature matrix, $\eta$ represents Gaussian noise, and $X$ denotes the perturbation level. The dataset is divided into three segments based on the stability ranking of nodes as determined by SAGMAN.

| Perturbation Level | KL divergence (bottom 20%) | KL divergence (mid 60%) | KL divergence (top 20%) |
|---|---|---|---|
| 0.4 | 0.01 | 0.03 | 0.03 |
| 0.8 | 0.09 | 0.16 | 0.19 |
| 1.2 | 0.43 | 0.56 | 0.59 |

## A.13 SAGMAN's Runtime Across Datasets

Figure 11 demonstrates the high efficiency of SAGMAN in processing large graphs, attributing its performance to the near-linear time complexity of its components.

## A.14 Hyperparameter Study of SAGMAN

In this section, we present a hyperparameter study of the SAGMAN framework to evaluate the impact of key parameters on its performance. Specifically, we analyze how varying the number of nearest neighbors $k$ used to construct the initial graph and the sparsification parameter *sparse_numer* affect the stability assessment and robustness of GNNs.

### A.14.1 Impact of $k$ and *sparse_numer*

The hyperparameter $k$ determines the number of nearest neighbors in the $k$-nearest neighbor (KNN) algorithm used to construct the initial graph for the manifold. The parameter *sparse_numer* controls the level of sparsification during graph pruning, influencing the density and connectivity of the resulting graph.

To assess the effects of these hyperparameters, we conducted experiments on the Cora dataset using a GCN model. We varied $k$ and *sparse_numer* while measuring the following metrics:

- **PGD Attack Accuracy (Stable Nodes)**: The classification accuracy on stable nodes after applying a PGD attack.
- **PGD Attack Accuracy (Unstable Nodes)**: The classification accuracy on unstable nodes after applying a PGD attack.
- **Cosine Similarity (Stable Nodes)**: The average cosine similarity between the original and perturbed embeddings of stable nodes.

- **Cosine Similarity (Unstable Nodes)**: The average cosine similarity between the original and perturbed embeddings of unstable nodes.

### A.14.2 RESULTS AND DISCUSSION

Table 12 summarizes the results of our experiments. We varied $k$ from 10 to 80 and *sparse_numer* from 2 to 5.

Table 12: Impact of $k$ and *sparse_numer* on SAGMAN's performance. 'NC' denotes configurations where ARPACK did not converge.

| $k$ | *sparse_numer* | PGD Acc. (Stable) | PGD Acc. (Unstable) | Cosine Sim. Diff. |
|-----|------|------|------|------|
| 10 | 2 | 1.00 | 0.74 | 0.99998 / 0.98302 |
| 10 | 3 | 1.00 | 0.96 | 0.99985 / 0.99983 |
| 10 | 4 | 0.89 | 1.00 | 0.99985 / 0.99987 |
| 30 | 2 | 1.00 | 0.74 | 0.99996 / 0.98636 |
| 30 | 3 | 1.00 | 0.81 | 0.99998 / 0.98579 |
| 30 | 4 | 1.00 | 0.74 | 0.99993 / 0.98528 |
| 50 | 3 | 1.00 | 0.81 | 0.99964 / 0.99986 |
| 50 | 4 | 0.93 | 0.81 | 0.99313 / 0.99965 |
| 50 | 5 | 1.00 | 0.89 | 0.99980 / 0.99979 |
| 80 | 2 | 0.96 | 0.89 | 0.99998 / 0.99722 |
| 80 | 3 | 0.96 | 0.81 | 0.99999 / 0.99809 |
| 80 | 4 | 1.00 | 0.74 | 0.99968 / 0.99678 |

**Effect of $k$:**   - When $k$ is small (e.g., $k = 10$), the initial graph captures local relationships but may miss some global structural information. - As $k$ increases to 30 and 50, the performance on stable nodes remains high, and the cosine similarity between original and perturbed embeddings for stable nodes remains close to 1.0, indicating robustness. - A larger $k$ (e.g., $k = 80$) does not necessarily lead to better performance and may cause computational challenges, as indicated by non-converging configurations (not shown in the table).

**Effect of *sparse_numer*:**   - Lower values of *sparse_numer* (e.g., 2 or 3) result in sparser graphs after pruning, which helps in maintaining high classification accuracy and robustness on stable nodes. - Increasing *sparse_numer* to 4 or 5 leads to denser graphs, which may capture more complex structures but can also introduce noise, potentially affecting stability. - Configurations with *sparse_numer* = 5 sometimes led to convergence issues during eigenvalue computations, suggesting that overly dense graphs may pose computational difficulties.

**Cosine Similarity Analysis:**   - The cosine similarity between the original and perturbed embeddings is consistently higher for stable nodes compared to unstable nodes. - For example, with $k = 10$ and *sparse_numer* = 2, the cosine similarity is 0.99998 for stable nodes and 0.98302 for unstable nodes, highlighting SAGMAN's ability to distinguish between stable and unstable nodes.

### A.15 ROBUSTNESS UNDER POISONING ATTACKS AND COMPARISON WITH LIPSCHITZ-BASED METHODS

In this section, we compare the robustness of SAGMAN with a Lipschitz-based stability method, specifically the LipReLU method (Jia et al., 2023), under poisoning attacks. While previous work (Jia et al., 2023) focuses exclusively on poisoning attacks, SAGMAN is primarily designed for evasion attacks. However, due to its versatility, SAGMAN can be deployed in poisoning scenarios and combined with other robustness techniques.

To demonstrate this, we conducted experiments using the DICE poisoning attack on the Cora dataset with a GCN model. We evaluated the robustness improvement provided by SAGMAN, LipReLU,

and a combination of both methods. The results highlight the accuracy improvement on the 10% most unstable samples, as identified by SAGMAN.

Table 13: Accuracy comparison under DICE poisoning attack for different numbers of edge perturbations. Best results are highlighted in **bold**.

| Method | 1 Edge | 10 Edges | 50 Edges | 100 Edges |
|---|---|---|---|---|
| LipReLU | 0.7862 | 0.7903 | 0.7782 | 0.7822 |
| SAGMAN | **0.8588** | 0.8427 | **0.8508** | **0.8548** |
| LipReLU + SAGMAN | 0.8468 | **0.8467** | 0.8467 | 0.8427 |

The results in Table 13 demonstrate that SAGMAN outperforms the Lipschitz-based method LipReLU in most cases, particularly as the number of edge perturbations increases. Combining SAGMAN with LipReLU also yields competitive results, indicating that SAGMAN can enhance the robustness provided by Lipschitz-based methods. This suggests that SAGMAN is a versatile plug-in that can be effectively integrated with existing robustness techniques to improve GNN resilience under poisoning attacks.

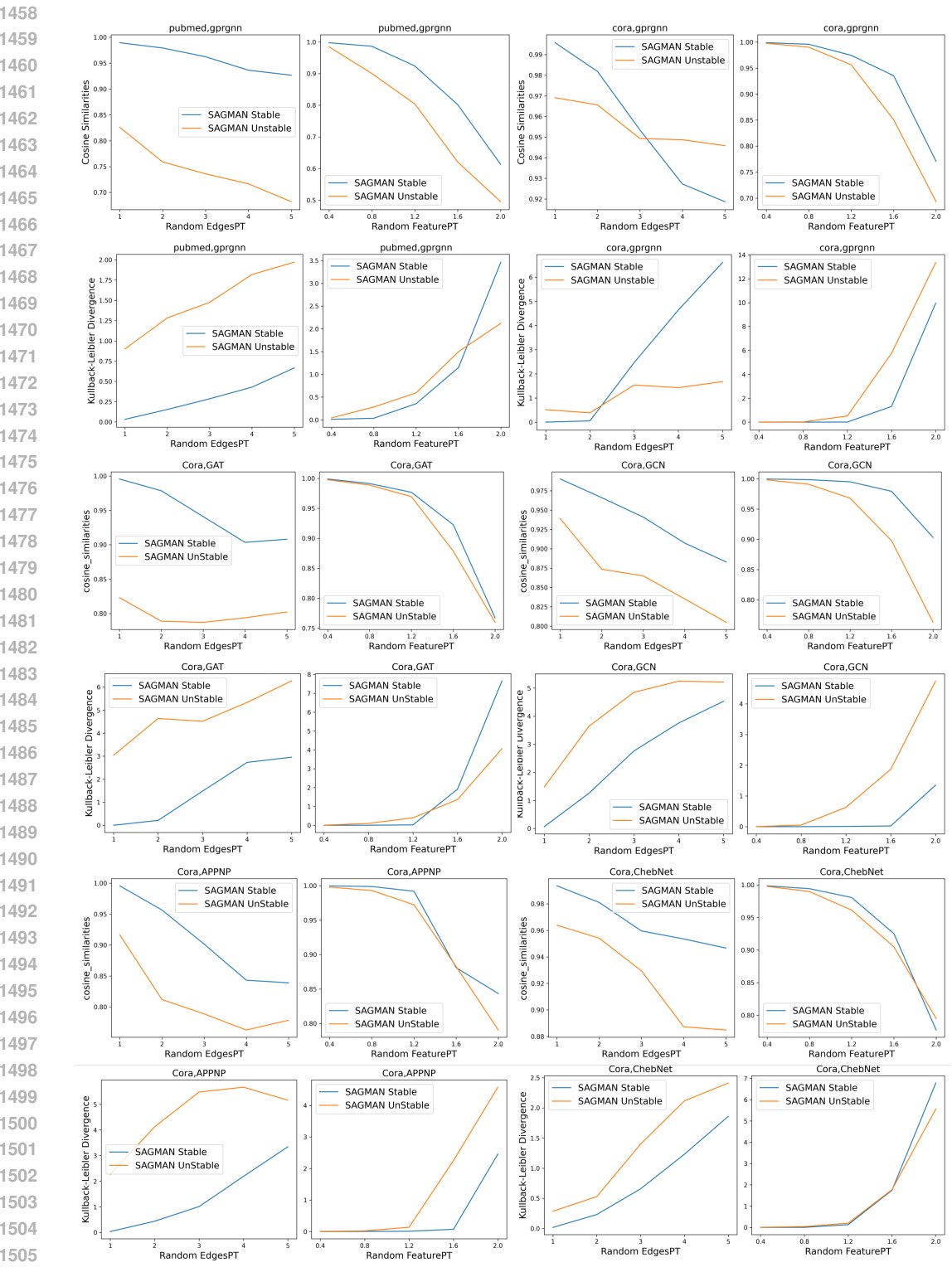

Figure 8: Figures represent cosine similarities and KL divergence. "Random EdgesPT" corresponds to the DICE edge evasion attack. "Random FeaturePT" refers to Gaussian noise evasion perturbation $X + \xi\eta$, where $X$ is feature matrix, $\eta$ is Gaussian noise, $\xi$ is noise level controls

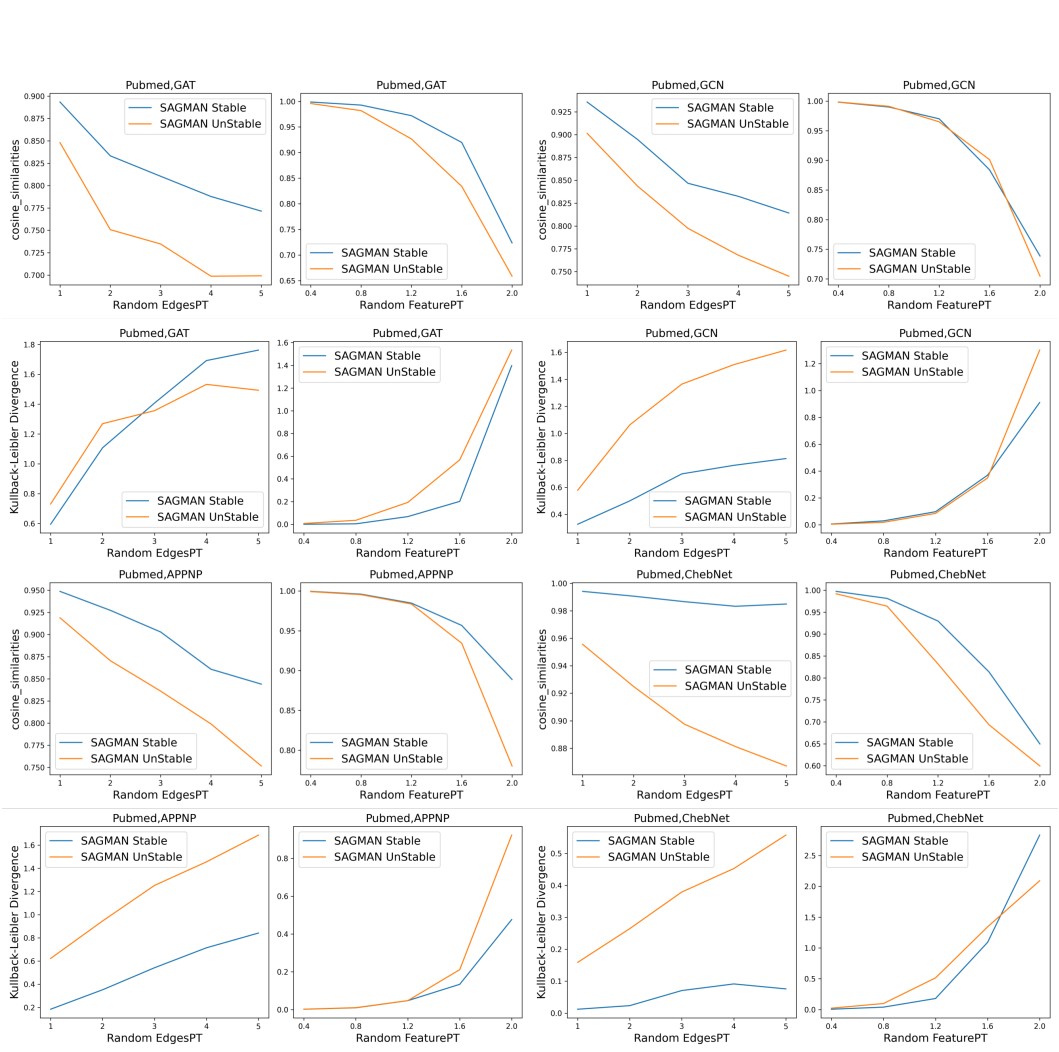

Figure 9: The horizontal axes, denoted by $X$, represent the magnitude of perturbation applied. 'Random EdgesPT' refers to the DICE adversarial attack scenario, in which pairs of nodes with different labels are connected and pairs with the same label are disconnected, with the number of pairs being equal to $X$. 'Random FeaturePT' indicates the application of Gaussian noise perturbation, expressed as $FM + X\eta$, where $FM$ denotes the feature matrix and $\eta$ represents Gaussian noise. The upper and lower subfigures illustrate the cosine similarity and the Kullback–Leibler Divergence (KLD). 'SAGMAN Stable/Unstable' denotes the samples that are classified as stable or unstable by SAGMAN, respectively.

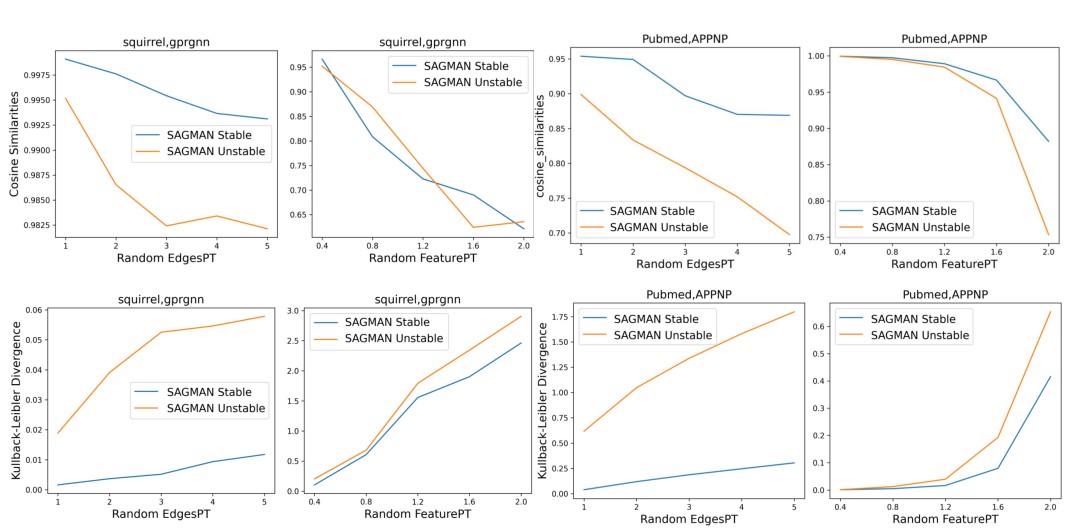

Figure 10: The horizontal axes, denoted by $X$, represent the magnitude of perturbation applied. 'Random EdgesPT' refers to the DICE adversarial attack scenario, in which pairs of nodes with different labels are connected and pairs with the same label are disconnected, with the number of pairs being equal to $X$. 'Random FeaturePT' indicates the application of Gaussian noise perturbation, expressed as $FM + X\eta$, where $FM$ denotes the feature matrix and $\eta$ represents Gaussian noise. The upper and lower subfigures illustrate the cosine similarity and the Kullback–Leibler Divergence (KLD). 'SAGMAN Stable/Unstable' denotes the samples that are classified as stable or unstable by SAGMAN, respectively.

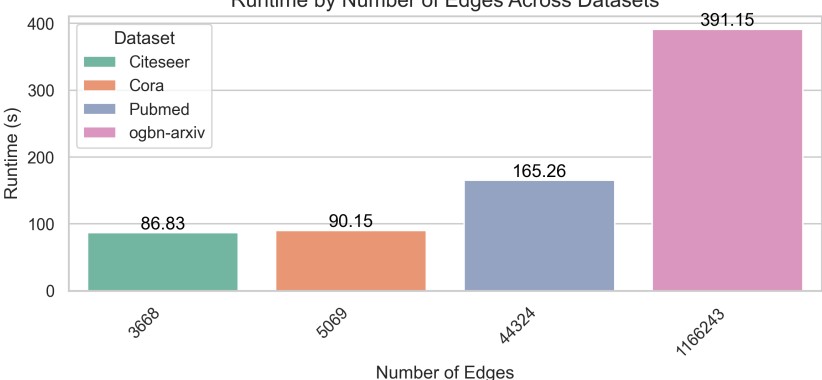

Figure 11: SAGMAN's runtime across datasets

