# OpenReview forum: "SAGMAN: Stability Analysis  of Graph Neural Networks  on the Manifolds"
_ICLR.cc/2025/Conference — ICLR 2025 Conference Withdrawn Submission_

### Official Review · Reviewer_UgfL · 2024-11-01

**Soundness:** 2
**Presentation:** 3
**Contribution:** 2
**Rating:** 3
**Confidence:** 3

**Summary:**

This paper introduces a framework for evaluating the GNNs' stability. Specifically, the metrics for defining the so-called DMD distance are selected as effective resistance distance. The initial graph is first embedded via its spectral characteristics, and a secondary graph is constructed by the KNN algorithm followed by the graph pruning (sparsification) approach. Hence, for any pair of nodes, e.g., p and q, their distance quotient on the embedded manifold is leveraged as the evaluation metric for measuring GNN (node level) stability. Experimental studies show promising results via graph data under adversarial attacks.

**Strengths:**

This paper well-organized the computational flow of inducing new DMD distance for evaluating GNN stability.

The newly proposed algorithm has nearly linear time complexity and empirical study shows the GNNs under the guidance of the proposed model own greater stability compared to their original counterparts.

**Weaknesses:**

Some of the contents are unclear, making the paper difficult to follow.

Some technical parts of the paper may need further justification.

**Questions:**

At the current stage, I have the following questions:

1. More content on how PGM acts on the graph data may be needed here. Also, if PGMs create a manifold for graphs, what is the metric of the manifold? Furthermore, does this (discrete) metric change over the GNN propagation?

2. When presenting the notation of PGM and the reason for preserving the resistance distance is essential, the introduction of stable or unstable nodes needs to be enriched in Appendix A.6; I couldn't find any formal definition or threshold of identifying what exactly is stable/unstable nodes.

3. I think the stability induced by the algorithm is limited to the GNNs that smooth the node features. What will be the conclusion/influence for those GNNs that typically fit the heterophilic graphs or can induce a sharpening dynamic to the node features? As in this case, each row of X will tend to be dissimilar to each other.

4. Perhaps a hyperparameter study for the algorithm with several components is necessary here.

5. Only to my view, as the proposed algorithm contains many components, they seems not be presented as a whole pipeline in the paper. Therefore, the author is suggested to refine the structure of the paper, e.g., adding some inter-contextual contents between each step.

Some minor changes are suggested as follows:
1. On row 234, the input embedding matrix $X = U_k$, I think $X$ should be the initial feature?

2. On row 759, the Appendix A.1 is self-refered.

---

> ### Author Response · Authors · 2024-11-25
>
> **Weakness 1**: *Some of the contents are unclear, making the paper difficult to follow.*
>
> We acknowledge that we cannot include every detail due to the page limitation. Could you please share more information regarding the unclear contents? In the revised version, we will refine these explanations.
>
> **Weakness 2**: *Some technical parts of the paper may need further justification.*
>
> We agree that additional justification would strengthen the technical sections. However, due to the page limitation, we provided more detailed explanations of the theoretical underpinnings of our approach in the Appendix. Could you please share more information on which part needs further justification?
>
> ---
>
> **Question 1**:
>
> In our framework, PGMs are used to learn the structure of graph-based manifolds from data embeddings. The metric we employ on these manifolds is the **effective resistance distance**, which captures both local and global structural information. This metric remains consistent during GNN propagation because it is computed based on the constructed manifolds prior to GNN processing. However, after GNN propagation, the output manifold (constructed from the GNN outputs) may exhibit changes in this metric due to the transformations applied by the GNN. By analyzing the changes between the input and output manifolds using DMD, we can effectively assess node-level robustness.
>
> **Question 2**:
>
> Thank you for highlighting this oversight. We define a node as **unstable** if small perturbations in the input (e.g. graph topology or node features) lead to significant changes in the output, indicated by a high Distance Mapping Distortion (DMD) value. Conversely, a **stable** node has a low DMD value, reflecting robustness to input variations. We will include a formal definition and specify thresholds (e.g., based on statistical deviations from the mean DMD value) used to categorize nodes, providing clear criteria for stability assessment.
>
> **Question 3**:
>
> You raise an important point. Our method is indeed most effective for GNNs that perform feature smoothing, common in homophilic graphs. However, for heterophilic graphs, we still observed SAGMAN distinguish stable and unstable nodes, as shown in Appendix Figure 10. We will discuss these considerations and potential extensions in the paper, acknowledging this limitation and proposing it as future work.
>
> **Question 4**:
>
> We agree that a hyperparameter study would provide valuable insights into the robustness and sensitivity of our method. Due to space constraints, we could not include this in the main paper. However, we will add a hyperparameter analysis in the supplementary material, examining the impact of key parameters such as the number of nearest neighbor\( k \), sparsification thresholds , and others. This will demonstrate how these parameters affect performance and guide practitioners in selecting appropriate values.
>
> **Question 5**:
>
> Thank you for this suggestion. We will refine the paper's structure by including a comprehensive flowchart or diagram that outlines the entire SAGMAN pipeline. Additionally, we will enhance the transitions between sections with interconnecting text that explains how each component leads to the next. This will provide a cohesive narrative and help readers understand how the components integrate into a unified framework.
>
> ---
>
> **Minor Comments**:
>
> - *On row 234, the input embedding matrix \( X = U_k \); I think \( X \) should be the initial feature?*
>
> As we mentioned in row 232, \( X = U_k \) represents the spectral embedding matrix derived from the graph's Laplacian eigenvectors, not the initial node features. We will clarify this in the text to prevent confusion, ensuring that the distinction between the initial features and the embedding matrix is clear.
>
> - *On row 759, the Appendix A.1 is self-referred.*
>
> We apologize for this error. We will correct the reference to point to the appropriate section, ensuring all cross-references are accurate.
>
> ---
>
> We hope that these clarifications address your concerns. Your feedback has been invaluable in helping us improve the clarity and quality of our paper. We will incorporate these revisions to enhance the presentation and technical depth. Thank you again for your thoughtful review.

---

> > ### Comment · Reviewer_UgfL · 2024-11-27
> > **Follow up Response**
> >
> > I thank the authors for your time and efforts on the rebuttal. The response has partially resolved my concern. I still have a few follow-up questions.
> >
> > 1. Followed by the question 3, I couldn't find any discussion and considerations in the revised manuscript, please let me know if I missed.
> >
> > 2. In the revised table 12, why sparse_numer are 3,4 and 5 for k =50, while others are 2,3 and 4?
> >
> > 3. I can see the description of the stable and unstable nodes added in the revised paper; that is, the stability of the node is measured by the DMD of the output of some models after inserting perturbation. The follow-up question is, did you fix the learnable part of the model every time you inserted the noise? If not, could you clarify how the effects of perturbations were isolated from changes due to the model's parameter updates?

---

> > > ### Author Response · Authors · 2024-11-28
> > >
> > > Dear Reviewer,
> > >
> > > Thank you for your continued engagement and thoughtful follow-up questions. We are glad to hear that our previous responses have partially resolved your concerns. Below, we address your remaining questions:
> > >
> > > **1. Discussion on GNNs for Heterophilic Graphs (Follow-up to Question 3)**
> > >
> > > In the revised manuscript, we added a discussion to address this concern. Specifically, on **Line 459**, we state:
> > >
> > > *"SAGMAN is most effective for GNNs that perform feature smoothing, common in homophilic graphs. However, for heterophilic graphs, we still observed that SAGMAN can distinguish between stable and unstable nodes, as shown in Figure 10."*
> > >
> > > **2. Variation in `sparse_numer` Values in Table 12**
> > >
> > > Thank you for pointing out the inconsistency in `sparse_numer` values for k=50 in Table 12. The variation arises due to computational limitations encountered during eigenvalue computations using the ARPACK library (https://docs.scipy.org/doc/scipy/tutorial/arpack.html). Specifically, for certain combinations of k and `sparse_numer`, we experienced convergence issues (`ARPACK error -1: No convergence`), which prevented us from obtaining reliable results.
> > >
> > > To ensure the validity of our experiments, we only included configurations where the computations successfully converged. For k=50, the converged results were obtained with `sparse_numer` values of 3, 4, and 5. In contrast, for other k values, the successful configurations were with `sparse_numer` values of 2, 3, and 4.
> > >
> > > We have clarified these convergence issues in the A14.2. Unless otherwise specified, all results in this paper use `k=50` and `sparse_numer = 4`.
> > >
> > > **3. Fixing Model Weights During Perturbation Experiments**
> > >
> > > Yes, we fixed the learnable parameters of the model during the perturbation experiments. Our goal was to ensure that any observed changes were due solely to the perturbations and not influenced by model parameter updates.
> > >
> > > Specifically, we:
> > >
> > > - **Trained the GNN model** to convergence on the original graph.
> > > - **Fixed the model weights** after training.
> > > - **Applied perturbations** to the input graph or node features for stable and unstable nodes separately.
> > > - **Measured the changes** in the model's output resulting from these perturbations.
> > >
> > > This approach allows for a fair comparison between stable and unstable nodes.
> > >
> > > ---
> > >
> > > We hope these responses fully address your questions. We are committed to making any further clarifications needed and appreciate your valuable feedback, which helps us improve the clarity and rigor of our work.
> > >
> > > Thank you again for your time and consideration.

---

### Official Review · Reviewer_ZBQU · 2024-11-03

**Soundness:** 3
**Presentation:** 3
**Contribution:** 2
**Rating:** 3
**Confidence:** 4

**Summary:**

The paper concerns the stability analysis of GNNs. The idea is inspired from the previously proposed Distance Mapping Distortion (DMD) metric. To match the requirement of having a near low-dimensional manifold for both input and output the authors have proposed to perform graph dimensionality reduction step for estimating the probabilistic graphical model. Additionally for scalability requirements the authors have also proposed spectral sparsification.

**Strengths:**

The paper studies an important problem of estimating the stability of GNNs
The framework is easy to understand and the computational complexity is linear w.r.t the graph dimensions.
Experimental evaluations validate the efficacy

**Weaknesses:**

The proposed framework although effective but largely uses a combination of many existing approaches.
The authors have clearly stated the distinction from the previous methods for analyzing GNN stability. However, the theoretical methods have guarantees. The current framework does not provide any strong theoretical guarantees.
Theoretical results related to DMD could be extended however the authors need to show theoretical guarantees on their approach of estimating the low-dimensional manifolds.
The authors have mentioned that previous literature lack analysis on stability towards feature perturbations. However, the results in  [R1] could be seen as stability bounds of GNN w.r.t. Feature perturbations. The authors are suggested to include it in the discussion.
The framework is restricted only to evasion attacks and does not considers a practical perturbation setup where the input graph itself is poisoned

**Questions:**

In addition to the above weaknesses:
How the framework performs against more sophisticated graph structure and attribute attacks (Metattack, PGD). Could you perform analysis on adaptive attacks [R2].
How the model performs on OGBN datasets.
Empirical comparison with Lipschitz-based stability methods

---

> ### Author Response · Authors · 2024-11-25
>
> ### **Weaknesses**
>
> 1. **Combination of Existing Approaches**
>
>     We appreciate the reviewer's feedback. As demonstrated in Section 3.3 and detailed in Appendix A.6, the previous DMD metric cannot be directly applied to GNN stability analysis, underscoring the necessity and uniqueness of the proposed SAGMAN framework.
>
>     The contribution is summarized as follows (as highlighted in Section 3): (1) SAGMAN introduces an innovative approach to convert the original input graph that may lie in a high-dimensional space into a low-dimensional graph-based manifold by leveraging PGMs. (2) To this end, an embedding matrix construction method and a scalable PGM construction approach are proposed, which will assure the preservation of graph resistance distances. (3) A novel spectral sparsification based on the short-cycle decomposition of weighted graphs is proposed to allow for the highly scalable construction of PGMs (low-dimensional graph-based manifolds).
>
>     A key contribution of our work lies in the synergistic integration of these components, which enables fine-grained, node-level stability analysis in GNNs—an aspect largely overlooked by prior approaches that focus on improving overall robustness.
>
> 2. **Lack of Strong Theoretical Guarantees**
>
>     In our work, we build upon theoretical results related to the DMD metric and effective resistance distances. The previous method [1] provides guarantees for recovering sparse graph structures preserving effective resistance distances. However, this method for learning PGMs may require numerous iterations to achieve convergence.
>
>     To address this, we propose a scalable PGM using spectral sparsification and graph decomposition, as detailed in Section 3.3. In Appendices A.4 and A.9, we provide theoretical analyses demonstrating that PGM can be obtained by our pruning strategy. These analyses offer strong theoretical guarantees on the accurate estimation of low-dimensional manifolds.
>
> 3. **Existing Stability Bounds for Feature Perturbations**
>
>     Could you please offer more information regarding the [R1] paper?
>
> 4. **Restriction to Evasion Attacks**
>
>     While previous work [2] focuses exclusively on poisoning attacks, SAGMAN specifically addresses evasion attacks. However, since SAGMAN is a versatile plug-in method, we can easily deploy SAGMAN into the poisoning attacks and combine with other robustness techniques.
>
>     To demonstrate this, we present the GCN robustness improvement results under the DICE poisoning attack, comparing SAGMAN with the previous SOTA method [2]. The results highlight the accuracy improvement on the 10% most unstable samples:
>
>     | DICE number of edge perturb | 1 | 10 | 50 | 100 |
>     | --- | --- | --- | --- | --- |
>     | LipReLU  | 0.7862 | 0.7903 | 0.7782 | 0.7822 |
>     | SAGMAN  | **0.8588** | 0.8427 | **0.8508** | **0.8548** |
>     | LipReLU +SAGMAN  | 0.8468 | **0.8467** | 0.8467 | 0.8427 |
>
> ### **Questions**
>
> 1. **Performance Against Sophisticated Attacks (Metattack, PGD)**
>
>     We would like to highlight that our evaluation already includes sophisticated attacks, such as Nettack. Also, we appreciate the reviewer’s suggestion to include more sophisticated attacks such as PGD. In response, we have conducted additional experiments using these attacks. Below, we present GAT case demonstrating the robustness  under PGD on cora:
>
>     | PGD Perturbation | **0.05** | **0.10** | **0.15** |
>     | --- | --- | --- | --- |
>     | Robust Accuracy | 1.0000 | 1.0000 | 1.0000 |
>     | Non-Robust Accuracy | 0.9630 | 0.8889 | 0.8148 |
>
>     Regarding Metattack, it is designed for **global attacks** that perturb the entire graph structure to degrade overall model performance. Since our framework focuses on **node-level stability analysis**, Metattack does not align well with our evaluation objectives. Metattack's global perturbations make it unsuitable for assessing the robustness of individual nodes in the context of our methodology.
>
> 2. **Analysis on Adaptive Attacks**
>
>     Could you please offer more information regarding the [R2] paper?
>
> 3. **Performance on OGBN Datasets**
>
>     Please see Figure 7 and Figure 11 in the Appendix for the OGBN results.
>
> 4. **Empirical Comparison with Lipschitz-based Stability Methods**
>
>     **Response:**
>
>     We compared the one Lipschitz method against poisoning attacks. Please refer to the table in **Weaknesses** No.4 above.
>
>
> [1] Feng, Zhuo. "SGL: Spectral graph learning from measurements." 2021 58th ACM/IEEE Design Automation Conference (DAC). IEEE, 2021.
>
> [2] Jia, Yaning, et al. "Enhancing node-level adversarial defenses by lipschitz regularization of graph neural networks." Proceedings of the 29th ACM SIGKDD Conference on Knowledge Discovery and Data Mining. 2023.

---

> > ### Comment · Reviewer_ZBQU · 2024-11-27
> > **References [R1] and [R2]**
> >
> > [R1]: Abbahaddou, Y., Ennadir, S., Lutzeyer, J. F., Vazirgiannis, M., & Boström, H. (2024). Bounding the Expected Robustness of Graph Neural Networks Subject to Node Feature Attacks.
> >
> > [R2]: Mujkanovic, F., Geisler, S., Günnemann, S., & Bojchevski, A. (2022). Are defenses for graph neural networks robust?.

---

> > > ### Author Response · Authors · 2024-11-28
> > >
> > > Dear Reviewer,
> > >
> > > Thank you for your thoughtful follow-up questions and for bringing these relevant references to our attention. We appreciate the opportunity to clarify and expand upon our work in light of your insights.
> > >
> > > ---
> > >
> > > **1. Relation to [R1]: Bounding the Expected Robustness of GNNs Subject to Node Feature Attacks**
> > >
> > > We acknowledge that [R1] introduces a theoretical framework for assessing the robustness of Graph Neural Networks (GNNs) against feature perturbations. Their work focuses on overall model robustness under feature attacks.
> > >
> > > However, Our work differs in several key aspects:
> > >
> > > - **Node-Level Stability Assessment**: While [R1] primarily addresses global robustness metrics, our approach focuses on **node-level** stability analysis. We employ the Distance Mapping Distortion (DMD) metric with effective resistance distances to quantify the stability of individual nodes. This allows us to rank nodes based on their susceptibility to perturbations, which is crucial for tasks like targeted defense and attack strategies.
> > > - **Spectral Manifold Approach**: We construct low-dimensional graph-based manifolds using spectral embeddings, preserving the effective resistance distances from the original graph.
> > > - **Practical Application and Scalability**: Our framework is designed to be computationally efficient for large graphs, using spectral sparsification and graph decomposition.
> > >
> > > Additionally, we attempted to reproduce the results [R1]. However, we and others have encountered difficulties in reproducing their results, as documented in a GitHub issue ([link](https://github.com/Sennadir/GCORN/issues/1)). This has limited our ability to provide a direct empirical result. Nevertheless, we acknowledge the importance of their theoretical contributions and will include a discussion of [R1] in our related work section to situate our approach within the broader context of GNN robustness research.
> > >
> > > ---
> > >
> > > **2. Analysis on Adaptive Attacks [R2]: Are Defenses for Graph Neural Networks Robust?**
> > >
> > > We recognize the significance of evaluating GNN defenses against adaptive attacks, where adversaries are aware of the defense mechanisms and tailor their strategies accordingly, as highlighted in [R2].
> > >
> > > In response to your suggestion, we conducted experiments to assess SAGMAN's effectiveness under adaptive attacks [R2].
> > >
> > > Our findings are as follows:
> > >
> > > | Model | Node Type | Clean Accuracy | Evasion Accuracy | Poisoned Accuracy |
> > > | --- | --- | --- | --- | --- |
> > > | **GNNGuard** | **Stable Nodes** | 1.0 | 0.625 | 0.625 |
> > > |  | **Unstable Nodes** | 0.833 | 0.0 | 0.125 |
> > > | **GCN** | **Stable Nodes** | 1.0 | 0.458 | 0.542 |
> > > |  | **Unstable Nodes** | 0.917 | 0.083 | 0.125 |
> > >
> > > We will include these findings in the revised manuscript to highlight SAGMAN's applicability in adaptive attack scenarios and its ability to bolster existing defense strategies against sophisticated adversaries.
> > >
> > > ---
> > >
> > > We appreciate your valuable feedback, which has helped us to strengthen our work and its presentation. We will incorporate discussions on [R1] and [R2] in our paper to provide a more comprehensive overview of related work and to position our contributions within the broader landscape of GNN robustness research.
> > >
> > > Thank you again for your time and consideration.
> > >
> > > ---
> > >
> > > **References:**
> > >
> > > [R1] Abbahaddou, Y., Ennadir, S., Lutzeyer, J. F., Vazirgiannis, M., & Boström, H. (2024). Bounding the Expected Robustness of Graph Neural Networks Subject to Node Feature Attacks.
> > >
> > > [R2] Mujkanovic, F., Geisler, S., Günnemann, S., & Bojchevski, A. (2022). Are Defenses for Graph Neural Networks Robust?

---

### Official Review · Reviewer_bLMT · 2024-11-03

**Soundness:** 3
**Presentation:** 3
**Contribution:** 3
**Rating:** 8
**Confidence:** 4

**Summary:**

The authors propose a new framework based on spectral theory, called SAGMAN, for stability analysis of GNNs. They analyze the distance distortions between the input graph manifold and output graph manifold as a measure of stability. To this end, the authors propose a distance preserving graph dimensionality reduction method to obtain a low dimensional input/output graph manifolds and use the DMD metric to characterize the stability. The authors show that this approach is scalable by showing that the time complexity is near-linear in the graph size. In the experimental section, the authors show that SAGMAN can be used in recommendation systems. Further, the authors show that this framework can be used to facilitate adversarial targeted attacks.

**Strengths:**

The paper is well-structured and well-written.

SAGMAN can be used to quantify stability at the node level, which I think is quite useful.

The paper has a significant theoretical component, backed up by some experimental data as well.

**Weaknesses:**

The paper evaluates SAGMAN mainly on node classification and recommendation tasks. Testing on other GNN tasks, like link prediction or community detection, could strengthen the generalizability of the findings​

SAGMAN needs to be compared with other robustness techniques to get a better idea of what specific advantages it offers versus what are the limitations

I think that ignoring d (which in theory can be O(|V|) for a complete graph) from the complexity analysis and claiming that the method is near-linear in time complexity in the size of the graph is slightly misleading.

**Questions:**

SAGMAN’s effectiveness depends on the input graph being well-represented in a low-dimensional space and having a significant eigengap. Is there any fix for this?

---

> ### Author Response · Authors · 2024-11-25
>
> **Weakness 1**:
>
> We appreciate your suggestion to assess the generalizability of SAGMAN across different GNN tasks.
> **Applicability to Link Prediction:**
>
> Link prediction primarily focuses on inferring missing edges or predicting future connections between nodes based on local neighborhood information and connectivity patterns. This task is inherently localized, relying heavily on immediate node proximities and the existence of common neighbors or local subgraph structures.
>
> SAGMAN, on the other hand, is designed to assess the stability of GNNs by capturing global structural properties through effective resistance distances, which consider the entire graph's topology. The metrics and methods we employ are optimized for scenarios where global connectivity and long-range interactions significantly impact the model's performance and stability.
>
> Applying SAGMAN to link prediction may not yield meaningful insights because:
>
> - **Local vs. Global Focus:** Link prediction's reliance on local structures means that global stability measures may not accurately reflect the task's requirements or the GNN's performance in this context.
> - **Distance Metrics:** The effective resistance distance used in SAGMAN captures global graph properties, which might obscure the local patterns critical for successful link prediction.
>
> Therefore, while theoretically possible, using SAGMAN for link prediction might not provide additional value and could potentially misrepresent the model's stability concerning this task.
>
> **Applicability to Community Detection:**
>
> Community detection involves partitioning the graph into clusters or communities based on densely connected nodes.
>
> While SAGMAN's global perspective might seem beneficial, the specific nature of community detection means that alternative methods tailored to detect and preserve community structures would be more appropriate.
>
> **Weakness 2**:
>
> We acknowledge the importance of comparing SAGMAN with existing robustness techniques. However, as you pointed out, while many studies focus on enhancing the robustness of GNN architectures, fewer have explicitly proposed methods that evaluate and rank node-level robustness. This scarcity makes direct comparisons challenging.
>
> Nevertheless, we can discuss the differences between SAGMAN and existing methods. Most robustness techniques aim to improve overall model robustness through architectural changes or training strategies, without providing fine-grained, node-level stability assessments. In contrast, SAGMAN offers a spectral framework to quantify stability at the individual node level, which can guide targeted interventions such as focused attacks or stability enhancements.
>
> In future revisions, we will include a discussion comparing SAGMAN conceptually with existing robustness methods, highlighting its unique contributions and potential integration with other techniques.
>
> **Weakness 3**:
>
> You raise a valid point regarding the complexity analysis. We agree that in the worst-case scenario of a complete graph, the average degree 'd' is O(|V|), making the time complexity O(|V|^2 m). However, most real-world graphs are sparse, with average degrees much smaller than the number of nodes. [1] In these cases, 'd' can be considered a constant or grows slowly with |V|, making the overall time complexity nearly linear in practice.
>
> To address this, we will revise the complexity analysis in the paper to explicitly account for 'd' and clarify that the near-linear time complexity holds for sparse graphs, which are common in practical applications.
>
> **Question**:
>
> Indeed, SAGMAN's effectiveness is tied to the graph's spectral properties, particularly the presence of a significant eigengap, which allows for meaningful dimensionality reduction while preserving structural information. For graphs that do not naturally exhibit significant eigengaps or are not well-represented in low-dimensional spaces, SAGMAN's performance may be limited.
>
> To address this limitation, one potential approach is to apply graph preprocessing techniques that enhance the spectral properties of the graph. For example, techniques like graph coarsening or spectral clustering can be used to restructure the graph into a form that has a more pronounced eigengap. Alternatively, augmenting the graph with additional edges or weights based on domain knowledge could improve its spectral characteristics.
>
> We acknowledge this as an important area for future research and plan to investigate methods to extend SAGMAN's applicability to a broader class of graphs.
>
> [1] Miao, Jianyu, et al. "Graph-based clustering via group sparsity and manifold regularization." IEEE Access 7 (2019): 172123-172135.

---

> > ### Comment · Reviewer_bLMT · 2024-11-26
> >
> > I thank the authors for their efforts and detailed answers. I have read the answers and I would like to stick to my score.

---

### Official Review · Reviewer_Fsow · 2024-11-05

**Soundness:** 2
**Presentation:** 2
**Contribution:** 2
**Rating:** 3
**Confidence:** 3

**Summary:**

This paper proposes a complicated framework called SAGMAN to measure the stability of trained GNN model.
It has three phases. 1. Construct the graph embedding based on node features and spectral features.
2. Use probabilistic graphical models (PGMs) to build low-dimension input graphs in the manifold.
	At the same time, the node embeddings of GNN outputs are also used to build low-dimension graphs in another manifold via PGM.
3. Use the  distance mapping distortion (DMD) calculations to measure the GNN stability.

**Strengths:**

It is interesting to find another metric to measure the GNN stability, and cooperating with manifold via PGMs is natural.

**Weaknesses:**

1. Graph robustness is a widely discussed problems.
However, a comprehensive discussion of related works is needed in this literature.
Moreover, more baselines and benchmarks are needed to be provided in the experimental part.
Survey paper [1] can help you gain more understanding about more recent works.
[1] A comprehensive survey on trustworthy graph neural networks: Privacy, robustness, fairness, and explainability.

2. For this metric of GNN stability, is there any conclusion from the experiments?
Does various graph robustness algorithms have been applied with this new metric?
Any comparison between this metric to other graph stability metric? For example, the prediction drop.

3. From my opinion, applying DMD on graphs is a problem as the manifold on graph is hard to define.
Although the PGMs can provide a solution, how good this PGMs can form the graph manifold is still a considerable questions.
Will the change of graph topology influence the mapping between graph structures and labels? This is also need to be analyzed.

**Questions:**

1. In Figure 1, do the three graphs represent internal views of the same input graph within the framework, or are they independent examples with no connection to each other?

2. How about the efficiency of the proposed algorithm?



-----
After rebuttal:
Thanks for your effort for the reply, and I will keep my score. My main concern remains the insufficient discussion and comparsion of related works in graph robustness area [1]. Thanks for your understanding.

[1] A comprehensive survey on trustworthy graph neural networks: Privacy, robustness, fairness, and explainability.

---

> ### Author Response · Authors · 2024-11-25
>
> Weakness 1: "Graph robustness is a widely discussed problem. However, a comprehensive discussion of related works is needed in this literature. Moreover, more baselines and benchmarks are needed to be provided in the experimental part. Survey paper [1] can help you gain more understanding about more recent works."
>
>  We agree that a more comprehensive discussion of related work would strengthen the paper. In the revised version, we will expand the related work section to include recent studies on graph robustness, particularly those highlighted in the survey paper [1] you mentioned. We acknowledge that while many studies focus on enhancing the robustness of GNN architectures, fewer have explicitly proposed methods for evaluating and ranking node-level robustness. This scarcity limits the availability of direct baselines for our experiments.
>
> **Weakness 2:** *"For this metric of GNN stability, is there any conclusion from the experiments? Have various graph robustness algorithms been applied with this new metric? Any comparison between this metric and other graph stability metrics? For example, the prediction drop."*
>
> Yes, our experiments yield several conclusions about the effectiveness of the SAGMAN metric:
>
> 1. **Effectiveness in Stability Assessment:** SAGMAN successfully distinguishes between stable and unstable nodes across multiple datasets and GNN architectures.
> 2. **Enhancing Adversarial Attacks:** Using SAGMAN to guide adversarial attacks results in higher error rates compared to traditional methods, indicating its utility in identifying vulnerable nodes.
> 3. **Enhancing defense regarding evasion attack:** Using SAGMAN significantly enhances defense against evasion attacks.
>
> Under perturbations, we have demonstrated in Table 5 that SAGMAN achieves higher accuracy than the SOTA GOOD-AT. Since the GOOD-AT paper does not report a prediction drop, we did not include it in our comparison. However, we provide the prediction accuracy for both approaches on the **unperturbed graph** here for reference.
>
> |  | Original | GOOD-AT | SAGMAN | SAGMAN+GOOD-AT |
> | --- | --- | --- | --- | --- |
> | cora_ml | 0.834 | 0.832 | 0.831 | 0.831 |
> | citeseer | 0.729 | 0.724 | 0.737 | 0.738 |
>
>
> **Weakness 3:** *"From my opinion, applying DMD on graphs is a problem as the manifold on graph is hard to define. Although the PGMs can provide a solution, how good this PGMs can form the graph manifold is still a considerable question. Will the change of graph topology influence the mapping between graph structures and labels? This also needs to be analyzed."*
>
> We acknowledge that defining manifolds on discrete graphs is challenging. To address this, we first embed the graph into a continuous space using spectral methods, specifically Laplacian Eigenmaps. This spectral embedding preserves essential structural properties of the original graph and approximates the **effective resistance distances.** Then we leverage PGMs to learn a graph structure that captures these dependencies from the spectral embeddings. By constructing PGMs from the embedding matrix Uk (as defined in Definition 3.1), we obtain a graph manifold that reflects both the data's inherent structure and the conditional dependencies.
>
> We understand your concern about whether changes in graph topology might affect the mapping between graph structures and labels. Our method focuses on preserving important structural features while simplifying the graph. By maintaining effective resistance distances through PGM construction, we retain both local and global connectivity patterns relevant to the labels.
>
>
> **Question 1:** *"In Figure 1, do the three graphs represent internal views of the same input graph within the framework, or are they independent examples with no connection to each other?"*
>
> Yes, it is the internal view. This sequential representation illustrates how the original input graph is transformed into low-dimensional (smooth) graph-based manifold through our framework to facilitate stability analysis.
>
> **Question 2:** *"How about the efficiency of the proposed algorithm?"*
>
> As section 3.6 introduced, the SAGMAN framework has a nearly linear time complexity. In our experiments in Appendix A.13, we demonstrate that SAGMAN can handle large datasets efficiently.
>
> We hope that our responses adequately address your concerns. We are committed to improving the paper based on your feedback and believe that the revisions will enhance the clarity and impact of our work.
>
> Thank you again for your thoughtful review.
>
> [1] A comprehensive survey on trustworthy graph neural networks: Privacy, robustness, fairness, and explainability.
>
> [2] Feng, Zhuo. "SGL: Spectral graph learning from measurements." 2021 58th ACM/IEEE Design Automation Conference (DAC). IEEE, 2021.

---

### Note · Authors · 2024-12-11

I have read and agree with the venue's withdrawal policy on behalf of myself and my co-authors.